# Scalable design of Error-Correcting Output Codes using Discrete Optimization with Graph Coloring

**Samarth Gupta**
Center for Computational Science & Engg.
Massachusetts Institute of Technology, USA
samarthg@mit.edu

**Saurabh Amin**
Laboratory for Information & Decision Systems
Massachusetts Institute of Technology, USA
amins@mit.edu

## Abstract

We study the problem of scalable design of Error-Correcting Output Codes (ECOC) for multi-class classification. Prior works on ECOC-based classifiers are limited to codebooks with small number of rows (classes) or columns, and do not provide optimality guarantees for the codebook design problem. We address these limitations by developing a codebook design approach based on a Mixed-Integer Quadratically Constrained Program (MIQCP). This discrete formulation is naturally suited for maximizing the error-correction capability of ECOC-based classifiers and incorporates various design criteria in a flexible manner. Our solution approach is tractable in that it incrementally increases the codebook size by adding columns to maximize the gain in error-correcting capability. In particular, we show that the maximal gain in error-correction can be upper bounded by solving a graph-coloring problem. As a result, we can efficiently generate near-optimal codebooks for very large problem instances. These codebooks provide competitive multi-class classification performance on small class datasets such as MNIST and CIFAR10. Moreover, by leveraging transfer-learned binary classifiers, we achieve better classification performance over transfer-learned multi-class CNNs on large class datasets such as CIFAR100, Caltech-101/256. Our results highlight the advantages of simple and modular ECOC-based classifiers in improving classification accuracy without the risk of overfitting.

## 1 Introduction

Error-correcting codes have found many applications in machine learning. Their use range from natural classification tasks [8, 24, 2, 17, 21] including multi-label classification [11] to robust classification [43, 46, 16] to federated learning [22] to zero-shot learning [40] to life-long learning [20]. The seminal paper of [8] proposed the use of error-correcting output codes (ECOCs) for classification tasks. Their framework lead to a significant activity on the design of ECOCs with a variety of features; for e.g. the use of different class of encodings [7, 17, 39], hierarchical classifiers [14, 12], decoding schemes [2, 9] and accounting for the underlying data-distribution [49, 47, 30, 33]. However, a key challenge that remains to be addressed is a flexible and tractable approach for designing high-quality codebooks. Our paper addresses this challenge.

We tackle some of the limitations of prior works on solving the *discrete* codebook design problem. In particular, we depart from the classical approach that performs a continuous relaxation of the discrete codebook design problem and attempts to solve the resulting non-linear optimization problem [7, 49, 47, 33]. That approach has been shown to scale to large number of classes $k$, but under the limitation that the size of the codebook (i.e. number of columns $L$) is small. However, with small $L$, codebooks do not fully exploit the error-correcting capability, which is the key motivation to use ECOCs in the first place. Besides, the continuous relaxation approach typically does not provide codebooks with strong optimality guarantees.

36th Conference on Neural Information Processing Systems (NeurIPS 2022).

We formulate the codebook design problem as a mixed integer quadratically constrained program (MIQCP) and develop a greedy algorithm to efficiently solve this MIQCP. Each iteration of our algorithm adds a new set of columns to the current codebook such that the gain in error-correcting capability is maximal. Importantly, we exploit a graph-coloring based upper-bound to accelerate the iterations of the greedy algorithm. Thus, our algorithm provides the flexibility to choose codebook size based on the desired level of final accuracy. The key advantage of our approach is that it easily scales to large problem instances ($k = 500$). In comparison, the recent paper of [16] generates high-quality codebooks but only for $k \leq 50$. Thus, our approach can be maximally leverage the benefit of ECOCs in aforementioned applications of error-correcting codes for *realistic* problem instances.

Our main contributions are as follows: (i) In contrast to [8, 33, 49, 47, 16], we model the codebook design problem as an element-wise optimization problem (MIQCP formulation). (ii) We develop a solution approach using a greedy algorithm to solve large instances of the MIQCP with low-optimality gaps. (iii) We make a theoretical connection to the vertex-coloring problem to upper bound the maximal gain in the error-correction which leads to faster iterations of the greedy algorithm. (iv) Our codebooks achieve very high classification accuracy similar to multi-class CNNs on datasets with small number of classes. (v) Finally, we show a novel benefit of ECOCs with transfer-learning on large class datasets. Our codebooks outperform multi-class CNNs when trained with transfer-learning under both nominal and robust feature settings.

## 2  ECOC for Classification

The ECOC-based framework [8], encodes each class of a $k$-class classification problem with a unique codeword of length $L$. This encoding results in a codebook (coding matrix) $\mathcal{M}$ of size $k \times L$. For binary codes: $\mathcal{M} \in \{+1, -1\}^{k \times L}$. The rows (resp. columns) of $\mathcal{M}$ correspond to distinct classes (resp. binary classifiers or *hypotheses*). Figure 1 shows two example codebooks for a classification problem with 5 classes. For every column in $\mathcal{M}$ a binary classifier is trained over the training data, where all training data from classes with entry $+1$ (resp. entry $-1$) forms the positive class (resp. the other class).

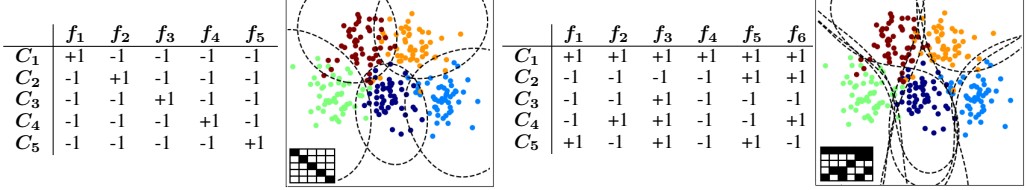

Figure 1: Example codebooks and their corresponding binary classifiers for a 5-class classification problem.

For a test example $\boldsymbol{x}$, let $f_1(\boldsymbol{x}), \ldots, f_L(\boldsymbol{x})$ denote the predicted class of each of the learned binary hypotheses, where $f_j(\boldsymbol{x}) \in \{+1, -1\} \ \forall \ j \in \{1, \ldots, L\}$. This encoding $\mathcal{F}(\boldsymbol{x})$, where $\mathcal{F}(\boldsymbol{x}) = [f_1(\boldsymbol{x}), \ldots, f_L(\boldsymbol{x})]$, is then associated with a class (i.e., a row of coding matrix $\mathcal{M}$), typically using a decoding scheme based on a similarity measure such as Hamming distance. A common approach is to compute the Hamming distances $d_H(\cdot, \cdot)$ between $\mathcal{F}(\boldsymbol{x})$ and each codeword $\mathcal{M}(i, \cdot) \ \forall i \in \{1, \ldots, k\}$ and predict the output class $\hat{y}$ as the class with the minimum distance:

$$d_H(\mathcal{M}(i, \cdot), \mathcal{F}(\boldsymbol{x})) := \sum_{j=1}^{L} \left( \frac{1 - \mathcal{M}(i, j) \times f_j(\boldsymbol{x})}{2} \right)$$

$$\hat{y} = \underset{i}{\operatorname{argmin}} \ d_H(\mathcal{M}(i, \cdot), \mathcal{F}(\boldsymbol{x})).$$

Importantly, the final classification performance of the above ECOC procedure depends on both the accuracy of the trained binary classifiers and the *error-correcting capability* of the codebook $\mathcal{M}$. In particular this capability increases with the Hamming distances between rows; i.e. a codebook with high pairwise Hamming distances across its rows has a higher error-correction capability.

**Definition 1.** *Hamming distance of a codebook (denoted as $\kappa_{\mathcal{M}}$) is defined as the minimum hamming distance between any two distinct pair of codewords (or rows) in $\mathcal{M}$.*
$$\kappa_{\mathcal{M}} = \min_{(i,j) \in \{1, \ldots k\}^2 | i < j} d_H(\mathcal{M}(i, \cdot), \mathcal{M}(j, \cdot))$$

**Proposition 1** (Error-Correction Capability). *A codebook $\mathcal{M}$ with hamming distance $\kappa_\mathcal{M}$, can always correct at-least $\lfloor \frac{\kappa_\mathcal{M}-1}{2} \rfloor$ errors.*

Proof of proposition 1 and all subsequent technical results are provided in supplementary information (SI).

## 3  Problem Formulation

For a $k$-class classification problem we want to find an optimal codebook of length $L$. Thus, we want to find the value of $k \times L$ entries of a codebook, where each $(i,j)$ entry (denoted as $x_{ij}$) can take values in $\{+1, -1\}$ such that the resulting codebook has high multi-class classification accuracy. To achieve high classification accuracy, we take the following design criteria [8, 16] into consideration:

1. The error-correcting property increases with the Hamming distance between rows; therefore it is desirable to have a high Hamming distance between any pair of rows. We ensure this by maximizing – the Hamming distance of the codebook (or the minimum of the row pairwise Hamming distances). For a binary codebook of size $k \times L$, an analytical upper bound on this objective is given by the well-known Plotkin's bound [37]: $\left\lfloor \frac{Lk}{2(k-1)} \right\rfloor$ .

2. The Hamming distance between any two pair of columns of the codebook must be at least 1 to avoid same columns and at most $k-1$ to avoid complementary columns. A good column separation is desirable in order to avoid correlations between the resulting hypotheses (or binary classifiers). Due to these reasons, we constrain the Hamming distances between any distinct pair of columns between $\rho_1$ and $\rho_2$, where $\rho_1 \geq 1$ and $\rho_2 \leq k-1$.

3. Finally we consider the balanced column criterion. This criterion allows to control the amount of imbalance in the resulting binary classification problem, i.e. the number of data-points in each of the two classes. This automatically ensures that no column has all entries as $+1$ or $-1$. Importantly, while balanced columns help in improving the error-correcting capability, imbalanced columns often result in easier binary classification problems, especially when working with linear classifiers, thus promoting higher multi-class accuracy [2, 9, 41]. In SI, we further discuss the nuances of shaping this trade-off in design of the codebook.

For notational convenience we denote the set of row indices as $\mathcal{N}$, the set of column indices $\mathcal{T}$, set of all pair of rows (or corresponding classes) $\mathcal{P}_\mathcal{N}$; and the set of all pair of columns $\mathcal{P}_\mathcal{T}$. Mathematically:

$$\mathcal{N} := \{1, \ldots, k\}, \; \mathcal{P}_\mathcal{N} := \left\{ (i, \hat{i}) \in \{1, \ldots, k\} \times \{1, \ldots, k\} \mid i < \hat{i} \right\},$$

$$\mathcal{T} := \{1, \ldots, L\}, \; \mathcal{P}_\mathcal{T} := \left\{ (j, \hat{j}) \in \{1, \ldots, L\} \times \{1, \ldots, L\} \mid j < \hat{j} \right\}.$$

We can now formulate the element-wise codebook design problem based on the above design criteria as the following Integer program ($\mathcal{IP}1$):

$$\mathcal{IP}1 : \max_x \; \min \{d_H^{1,2}(x), \; d_H^{1,3}(x), \; \ldots, \; d_H^{k-1,k}(x)\} \tag{1a}$$

$$\text{s.t.} \quad d_H^{i,\hat{i}}(x) = \frac{1}{2}\Big(L - \sum_{j=1}^{L} x_{ij} \times x_{\hat{i}j}\Big) \qquad \forall \, (i, \hat{i}) \in \mathcal{P}_\mathcal{N} \tag{1b}$$

$$\rho_1 \leq \frac{1}{2}\Big(k - \sum_{i=1}^{k} x_{ij} \times x_{i\hat{j}}\Big) \leq \rho_2 \qquad \forall \, (j, \hat{j}) \in \mathcal{P}_\mathcal{T} \tag{1c}$$

$$-\gamma \leq \sum_{i=1}^{k} n_i x_{ij} \leq \gamma \qquad \forall \, j \in \mathcal{T} \tag{1d}$$

$$x_{ij} \in \{+1, -1\} \qquad \forall \, (i, j) \in \mathcal{N} \times \mathcal{T} \tag{1e}$$

The objective of $\mathcal{IP}1$ is to maximize the minimum Hamming distances between different pairs of rows (1a). Constraint (1b) computes the Hamming distance between rows; (1c) ensures that column separation lies in the desirable range $[\rho_1, \rho_2]$; (1d) ensures that every column results in a valid balanced binary classification problem, where $n_i$ is the number of training samples in $i$-th class of the dataset, while $\gamma$ controls the degree of allowed imbalance; and finally (1e) ensures that $x_{ij}$ takes integer values in $\{+1, -1\}$. The max-min objective (1a) can be simplified by introducing an auxiliary variable $t$, where $t := \min \{d_H^{1,2}(x), \; d_H^{1,3}(x), \; \ldots, \; d_H^{k-1,k}(x)\}$ and adding constraints $t \leq d_H^{1,2}(x), \ldots, t \leq d_H^{k-1,k}(x)$.

Note that (1b) and (1c) contain bilinear terms and are non-convex in nature. Thus $\mathcal{IP}1$ can be categorized as a non-convex Mixed Integer Quadratic Constrained Program (MIQCP). $\mathcal{IP}1$ cannot be solved directly as (1b) and (1c) contains bilinear terms and therefore these constraints are linearized to get a linear-relaxation of $\mathcal{IP}1$. Each bi-linear term is replaced with an auxiliary variable $z_{ijpq}$, where $z_{ijpq} \coloneqq x_{ij}x_{pq}$. Additional linear constraints known as McCormick-inequalities [34] are then added to lower and upper bound $z_{ijpq}$. Further details about McCormick inequalities (including their derivation) are provided in supplementary information (SI). For the bilinear constraint $z_{ijpq} = x_{ij}x_{pq}$, these inequalities are given by (2a) and (2b).

$$\text{Lower:} \quad z_{ijpq} \geq -x_{ij} - x_{pq} - 1; \ z_{ijpq} \geq \ x_{ij} + x_{pq} - 1 \tag{2a}$$

$$\text{Upper:} \quad z_{ijpq} \leq \ x_{ij} - x_{pq} + 1; \ z_{ijpq} \leq -x_{ij} + x_{pq} + 1 \tag{2b}$$

Also, (2a) and (2b) respectively provide convex lower and upper envelopes to the feasible set admitted by the bilinear constraint [1].

**Lemma 1.** *McCormick relaxation of $\mathcal{IP}1$, denoted as $\mathcal{MC}(\mathcal{IP}1)$, is tight:*
$\left\{ (z_{ijpq}, x_{ij}, x_{pq}) \in \mathbb{R} \times \{+1, -1\}^2 \,|\, z_{ijpq} = x_{ij}x_{pq} \right\} \equiv \left\{ (z_{ijpq}, x_{ij}, x_{pq}) \in \mathbb{R} \times \{+1, -1\}^2 \,|\, (2a), (2b) \right\}.$

Thanks to lemma 1, $\mathcal{IP}1$ can now be solved by solving the MILP (i.e. $\mathcal{MC}(\mathcal{IP}1)$) obtained after replacing the bilinear terms in (1b)-(1c) with the corresponding McCormick inequalities. Note that $\mathcal{IP}1$ has $L \times |\mathcal{P}_\mathcal{N}| + k \times |\mathcal{P}_\mathcal{T}| = L \times \binom{k}{2} + k \times \binom{L}{2}$ number of bilinear terms. If $L \approx k$, $\mathcal{IP}1$ will have $\mathcal{O}(k^3)$ number of bilinear terms. However in practice, linear programming (LP)-relaxation of $\mathcal{MC}(\mathcal{IP}1)$ can be quite loose and hence even with the tight reformulation, $\mathcal{IP}1$ cannot be used to tractably solve the optimal codebook design problem (especially when $L$ is large).

Therefore, instead of generating optimal codebooks by solving $\mathcal{IP}1$ directly in one-shot, we next develop a scalable greedy solution approach to generate good-quality (near optimal) codebooks for large $k$ and $L$.

## 4 Proposed Solution Approach

In this section we develop an iterative greedy algorithm to generate near-optimal codebooks i.e. codebooks with low-optimality gap. To develop our approach, we utilize the following monotonicity result of lemma 2.

**Lemma 2.** *Let $\mathcal{M}$ and $\tilde{\mathcal{M}}$ be two binary error-correcting code of size $k \times l$ and $k \times (l + \tilde{l})$ respectively, where $\tilde{l} \geq 0$, such that all columns in $\mathcal{M}$ are also in $\tilde{\mathcal{M}}$, i.e. $\mathcal{M} \subseteq \tilde{\mathcal{M}}$. Then, the Hamming distances (recall 1) of $\mathcal{M}$ and $\tilde{\mathcal{M}}$ satisfy: $\kappa_\mathcal{M} \leq \kappa_{\tilde{\mathcal{M}}}$, implying that the error-correcting capability of $\tilde{\mathcal{M}}$ is at least as good as $\mathcal{M}$.*

This intuitive result serves as the starting point of our solution approach. In particular, from Lemma 2, we know that the error correcting capability of a codebook $\mathcal{M}$ of size $k \times l$ can be improved further by increasing the number of columns, i.e. adding $\tilde{l}$ columns. Naturally, this improvement not only depends on $\tilde{l}$, but also on how the entries of the new columns are selected. Our solution approach progressively adds new columns to an existing codebook by solving a smaller integer program, denoted $\mathcal{IP}2$; see figure 2. This integer program maximizes the hamming distance of the resulting codebook (i.e. its error-correcting capability), while ensuring that desired column separation for *all* (i.e. new and old columns) is achieved and the resulting codebook still remains a valid binary error-correcting code with balanced columns.

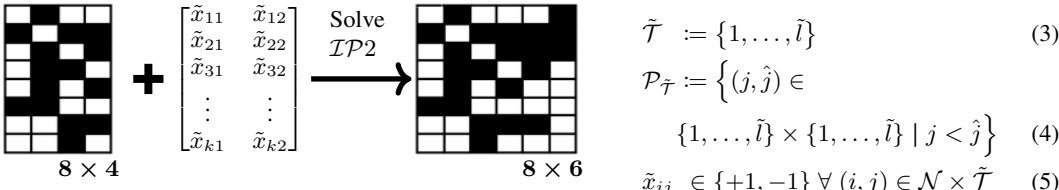

$$\tilde{\mathcal{T}} \coloneqq \{1, \dots, \tilde{l}\} \tag{3}$$

$$\mathcal{P}_{\tilde{\mathcal{T}}} \coloneqq \Big\{ (j, \hat{j}) \in$$

$$\{1, \dots, \tilde{l}\} \times \{1, \dots, \tilde{l}\} \,|\, j < \hat{j} \Big\} \tag{4}$$

$$\tilde{x}_{ij} \in \{+1, -1\} \ \forall \ (i, j) \in \mathcal{N} \times \tilde{\mathcal{T}} \tag{5}$$

Figure 2: Adding new columns to an existing codebook

Before proceeding, we introduce new notation for mathematical convenience. Analogous to $\mathcal{T}$, (resp. $\mathcal{P}_\mathcal{T}$) let $\tilde{\mathcal{T}}$ denote the set of new columns (resp. $\mathcal{P}_{\tilde{\mathcal{T}}}$ denote the set of pair of new columns). Furthermore, let the entries of the new columns be denoted as $\tilde{x}_{ij}$. Mathematically these are defined in (3), (4) and (5) respectively. We are ready to present $\mathcal{IP}2$ which takes a valid codebook $\mathcal{M}$ of size $k \times l$ and number of columns $\tilde{l}$ to be added as inputs.

$$\mathcal{IP}2 : \max_{\tilde{x}} \ \min \ \{d_H^{1,2}(\tilde{x}), \ d_H^{1,3}(\tilde{x}), \ \ldots, \ d_H^{k-1,k}(\tilde{x})\} \tag{6a}$$

$$d_H^{i,\hat{i}}(\tilde{x}) = \frac{1}{2}\left(l + \tilde{l} - \sum_{r=1}^{l} \mathcal{M}(i,r) \times \mathcal{M}(\hat{i},r) \ + \sum_{j=1}^{\tilde{l}} \tilde{x}_{ij} \times \tilde{x}_{\hat{i}j}\right) \qquad \forall \, (i,\hat{i}) \in \mathcal{P}_{\mathcal{N}} \tag{6b}$$

$$\rho_1 \le \frac{1}{2}\left(k - \sum_{i=1}^{k} \tilde{x}_{ij} \times \tilde{x}_{i\hat{j}}\right) \le \rho_2 \qquad \forall \, (j,\hat{j}) \in \mathcal{P}_{\tilde{\mathcal{T}}} \tag{6c}$$

$$\rho_1 \le \frac{1}{2}\left(k - \sum_{i=1}^{k} \mathcal{M}(i,j) \times \tilde{x}_{i\hat{j}}\right) \le \rho_2 \qquad \forall \, (j,\hat{j}) \in \{1,\ldots,l\} \times \tilde{\mathcal{T}} \tag{6d}$$

$$-\gamma \le \sum_{i=1}^{k} n_i \tilde{x}_{ij} \le \gamma \qquad \forall \, j \in \tilde{\mathcal{T}} \tag{6e}$$

$$\tilde{x}_{i,j} = \{+1,-1\} \qquad \forall \, (i,j) \in \mathcal{N} \times \tilde{\mathcal{T}} \tag{6f}$$

The objective function (6a) maximizes the the error-correcting capability. Constraint (6b) computes the Hamming distance between pairs of rows (or classes) in the resulting codebook; (6c) ensures desired column separation between any pair of *new* columns; (6d) ensured desired separation between new columns and existing columns in input $\mathcal{M}$; (6e) ensures that new columns are balanced and (6f) ensures that the new codebook is a binary codebook.

Notice the key difference between $\mathcal{IP}2$ and $\mathcal{IP}1$: since $\tilde{l} \ll l$, the constraints (6d), which account for the majority of columns separation conditions in the resulting codebook are *linear* in $\tilde{x}$ as the entries of $\mathcal{M}$ are known. Further, the total number of bi-linear terms in row-separation constraints (6b) and column separation (between new columns) constraints (6c) are much smaller in number. These features of $\mathcal{IP}2$ makes it solvable even for large $k$ by off-the-shelf IP solvers which typically employ branch-and-bound (B&B) procedure.

It turns out that we can pre-compute a good quality (potentially tight) upper-bound to $\mathcal{IP}2$. In section 4.1, we explore two different ways to generate an upper bound to $\mathcal{IP}2$. Our computational results show that a good upper bound can accelerate the termination of the B&B procedure for $\mathcal{IP}2$. Since our greedy algorithm 4.2 relies on solving $\mathcal{IP}2$ repeatedly to increase the error-correcting capability, the overall computational gain can be significant.

### 4.1 Upper Bound to $\mathcal{IP}2$

The integer constraint (6f), $\tilde{x}_{ij} \in \{+1,-1\}$ in $\mathcal{IP}2$ can be equivalently re-written as $\tilde{x}_{ij}^2 = 1$ and the resulting problem can be categorized as a *continuous*, non-convex quadratic program. We can generate an upper bound by taking the Rank-1 semi-definite programming (SDP) relaxation of this quadratic program (for details see SI). The tightness of this relaxation is same as taking the dual of aforementioned quadratic program (equivalently $\mathcal{IP}2$), which is also a SDP and is dual to the Rank-1 SDP relaxation. However, solving the SDP-relaxation can be expensive for large $k$ even with modern first-order SDP cone solvers (for eg: SCS [36]). More importantly our computational experiments suggest that the SDP-relaxation does not provide a good quality upper-bound to $\mathcal{IP}2$.

We now discuss a second procedure to generate an upper bound to $\mathcal{IP}2$. To begin with, note that the addition of $\tilde{l}$ columns to an existing codebook $\mathcal{M}$, as done by $\mathcal{IP}2$, can be equivalently viewed as the appending codewords (denoted as $c_i$) of length $\tilde{l}$ to each row $i$ of the codebook $\mathcal{M}$, see figure 3. For binary codes there are $2^{\tilde{l}}$ possible distinct codewords of length $\tilde{l}$, let the set of these codewords be denoted as $\mathcal{C}(\tilde{l})$. Essentially in solving $\mathcal{IP}2$, we seek to choose codewords from $\mathcal{C}(\tilde{l})$, such that the codebook resulting from appending them to the existing rows of $\mathcal{M}$ maximally increases the error-correcting capability. In particular, out of all $\binom{k}{2}$ row pairs, $\mathcal{IP}2$ tries to increase the hamming distance between row pairs which are at the minimum in the existing codebook, as these minimum hamming distance row pairs decide the error-correcting capability (recall proposition 1).

For a given codebook $\mathcal{M}$, we define the set of minimum distance row-pairs (denoted $\mathcal{E}_{\mathcal{M}}$) as:
$$\mathcal{E}_{\mathcal{M}} = \left\{(i,j) \in \{1,\ldots,k\} \times \{1,\ldots,k\} \mid i < j \text{ and } d_H(\mathcal{M}(i,\cdot), \mathcal{M}(j,\cdot)) = \kappa_{\mathcal{M}}\right\} \tag{7}$$

For example, the minimum hamming distance between any two distinct pair of rows in the codebook shown in Figure 3 is 4, and there are multiple row pairs for which the Hamming distance is 4. In

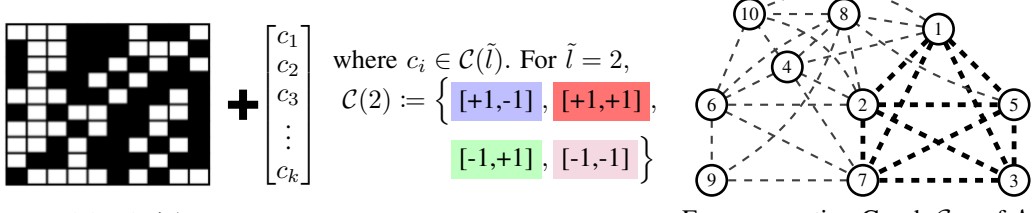

Codebook $\mathcal{M}$                               Error-correcting Graph $\mathcal{G}_{\mathcal{M}}$ of $\mathcal{M}$

Figure 3: Codebook $\mathcal{M}$ (left) and its corresponding error-correcting graph $\mathcal{G}_{\mathcal{M}}$ (right). Since $\mathcal{G}_{\mathcal{M}}$ has a clique of size 5 (shown in bold edges), therefore chromatic number $\xi(\mathcal{G}_{\mathcal{M}}) \geq 5$, thus $\mathcal{G}_{\mathcal{M}}$ cannot be colored with four colors (or codewords in $\mathcal{C}(2)$). Therefore adding codewords of size 2 will not result in improving the hamming distance of $\mathcal{M}$.

general, for a codebook $\mathcal{M}$ of size $k \times L$, we can represent the set $\mathcal{E}_{\mathcal{M}}$ as an undirected graph (denoted $\mathcal{G}_{\mathcal{M}}$) with $k$-vertices, where vertex $i$ corresponds to row $i$ in $\mathcal{M}$ $\forall\, i \in \{1, \ldots, k\}$. For every $(i, j)$ in $\mathcal{E}_{\mathcal{M}}$, an edge is present between vertices $i$ and $j$ in $\mathcal{G}_{\mathcal{M}}$. We refer to graph $\mathcal{G}_{\mathcal{M}}$ as the *error-correcting graph* of $\mathcal{M}$. Mathematically,

$$\mathcal{G}_{\mathcal{M}} = (\mathcal{V}, \mathcal{E}_{\mathcal{M}}), \text{ where } \mathcal{V} = \{1, \ldots, k\}$$

Figure 3 illustrates a codebook $\mathcal{M}$ and its corresponding error-correcting graph $\mathcal{G}_{\mathcal{M}}$. This viewpoint enables us to make two important observations:

1. Increasing the hamming distance of row-pairs in $\mathcal{E}_{\mathcal{M}}$ by at least 1 through addition of columns (or equivalently by appending codewords $c_i$) to $\mathcal{M}$, effectively increases the minimum hamming distance of the entire codebook by at least 1.

2. Any pair of distinct codewords in the set $\mathcal{C}(\tilde{l})$ differ by at least 1 and at most $\tilde{l}$, i.e. :
$$1 \leq d_H(c_i, c_j) \leq \tilde{l} \qquad\qquad \forall\, c_i, c_j \,\in\, \mathcal{C}(\tilde{l}),\ i \neq j.$$

These observations lead to the following claim in proposition 2:

**Proposition 2.** *For a codebook $\mathcal{M}$, if there exists an assignment of codeword $c_i \in \mathcal{C}(\tilde{l})$ to each vertex of graph $\mathcal{G}_{\mathcal{M}}$ such that no two connected vertices receive the same codeword, then the hamming distance of the codebook $\kappa_{\mathcal{M}}$ can be increased by at least 1 by adding codewords from $\mathcal{C}(\tilde{l})$ to the rows of $\mathcal{M}$.*

In fact the condition specified by proposition 2, can be easily verified by solving an instance of the graph-coloring problem on $\mathcal{G}_{\mathcal{M}}$, where each codeword $c_i \in \mathcal{C}(\tilde{l})$ can be viewed as a unique color. Before proceeding further, we recall some useful definitions related to graph coloring. A *proper-vertex coloring* is an assignment of colors to the vertices of a graph so that no two adjacent vertices have the same color.

**Definition 2** (Chromatic number). *The chromatic number of a graph $\mathcal{G}$, denoted as $\xi(\mathcal{G})$ is the minimum number of colors required for a proper vertex coloring of the graph $\mathcal{G}$.*

**Theorem 1.** *For any binary code $\tilde{\mathcal{M}}$ resulting from adding $\tilde{l}$ columns to an existing binary code $\mathcal{M}$, the following holds regarding the hamming distance (recall 1) $\kappa_{\tilde{\mathcal{M}}}$ of the code $\tilde{\mathcal{M}}$:*

- *If the chromatic number of the graph $\mathcal{G}_{\mathcal{M}}$, $\xi(\mathcal{G}_{\mathcal{M}})$ is greater than the size of the set of possible codewords $\mathcal{C}(\tilde{l})$, i.e. $\xi(\mathcal{G}_{\mathcal{M}}) > |\mathcal{C}(\tilde{l})|$, then $\kappa_{\tilde{\mathcal{M}}} = \kappa_{\mathcal{M}}$.*

- *In particular, for $\tilde{l} \in \{1, 2\}^{*}$ :*

$$\kappa_{\tilde{\mathcal{M}}} \leq \begin{cases} \kappa_{\mathcal{M}} + \tilde{l}, & \text{if } \xi(\mathcal{G}_{\mathcal{M}}) = 2 \\ \kappa_{\mathcal{M}} + \tilde{l} - 1, & \text{if } 3 \leq \xi(\mathcal{G}_{\mathcal{M}}) \leq 4 \\ \kappa_{\mathcal{M}}. & \text{if } \xi(\mathcal{G}_{\mathcal{M}}) \geq 5 \end{cases}$$

**Corollary 2.** *An upper bound to $\mathcal{IP}2$ is provided by theorem 1 or $\mathcal{IP}2$ is infeasible.*

Since for a given $\mathcal{M}$, theorem 1 provides a valid upper-bound to the hamming distance (or equivalently the error-correcting capability) of all codebooks resulting from adding $\tilde{l}$ columns to $\mathcal{M}$, therefore it

---
*Extension to cases where $\tilde{l} \geq 3$ is provided in SI.

| Algorithm 1 Greedy |
| --- |

1: Generate first column $C_1$ randomly.
2: $\mathcal{M} \leftarrow C_1$ and start counter: $i = 1$
3: **while** $i < L$ **do**
4:     Compute $\xi(\mathcal{G}_{\mathcal{M}})$.
5:     Compute upper bound to $\mathcal{IP}2$ using $\xi(\mathcal{G}_{\mathcal{M}})$ and $\tilde{l}$.
6:     Solve $\mathcal{IP}2$ with latest $\mathcal{M}, \tilde{l}$ and upper-bound.
7:     Update $\mathcal{M}$ with the solution of $\mathcal{IP}2$
8:     $i = i + \tilde{l}$
9: **end while**
10: **return** $\mathcal{M}$

| $k$ | Time (in secs.) |
| --- | --- |
| 50 | 0.063 |
| 100 | 0.148 |
| 150 | 0.099 |
| 200 | 0.165 |
| 250 | 1.039 |
| 300 | 0.234 |
| 350 | 0.134 |
| 400 | 1.921 |
| 450 | 0.813 |
| 500 | 0.852 |

Table 1: Avg. time taken to compute the chromatic number $\xi(\mathcal{G}_{\mathcal{M}})$ for different $k$.

automatically also provides an upper-bound to $\mathcal{IP}2$. Further, the color-assignment can be used to generate a (partial) feasible solution to $\mathcal{IP}2$. Technical details of feasible solution generation are provided in SI.

Note that the problem of computing the chromatic number of a graph (or analogously solving the vertex-coloring problem) has been extensively studied in the literature. This includes heuristics such as DSATUR [4] and RLF [28], exact backtracking algorithms [26] and more recent integer programming based exact methods [18, 35, 32]. We solve our graph-coloring problem by formulating and solving it as an integer program (IP); please refer to SI for details. In table 1, we report the average time needed to solve the vertex-coloring problem for various error-correcting graphs $\mathcal{G}_{\mathcal{M}}$ with different $k$. In almost all cases we are able to solve the graph-coloring problem extremely fast within 1-2 seconds. This is mainly due to the highly sparse structure of $\mathcal{G}_{\mathcal{M}}$ as large number of nodes tend to have low or even zero-degree resulting from high diversity in row pair-wise Hamming distances.

### 4.2 Greedy Algorithm

We are now ready to present our Greedy Approach (Algorithm 1) for generating a codebook of size $k \times L$ which is a near-optimal feasible solution to our original codebook design problem. Initially, we randomly pick the first column of the codebook such that it is a valid balanced binary classifier. Then, a second column is added by solving $\mathcal{IP}2$ with $\tilde{l} = 1$, this provides us with a codebook of size $k \times 2$ that satisfies all the design criteria. With this as our current codebook with two columns, we then continue to append more columns by again solving $\mathcal{IP}2$. This procedure is continued until we obtain a codebook with $L$ columns.

Practically, it is important to keep $\tilde{l}$ reasonably small so that the solution to $\mathcal{IP}2$ in each iteration of Algorithm 1 is obtained quickly. In the first few iterations, since $\mathcal{M}$ is small, therefore the number of column-separation constraints between the existing and new columns (6d) is small. Thus, for large $k$, i.e. $k > 50$, for first $\lceil \log_2 k \rceil^{\dagger}$ iterations we solve $\mathcal{IP}2$ with $\tilde{l} = 1$ and use $\tilde{l} \geq 2$ in the subsequent iterations.

### 4.3 Flexibility in choosing the size of the codebook $L$

For most ECOC design approaches in literature, one needs to pre-define $L$ to solve the codebook-design and then train each of the binary classifiers. If the resulting multi-class performance is not satisfactory then the process is repeated: increase $L$, resolve the codebook design problem and re-train all the columns in the new codebook. Naturally, this iterative design process is quite burdensome because of the computation effort spent on solving and training codebooks in each iteration.

In a sharp contrast, our design approach does not require pre-specifying $L$ as columns can be greedily added to build codebooks. In particular, we can start off with a relatively small $L$, train the resulting columns and evaluate the multi-class performance. If required, we can then append more columns to the existing codebook by solving $\mathcal{IP}2$, train the new columns and re-evaluate the multi-class performance. This procedure can be repeated until minimal or no improvement is observed. Importantly the computation effort spent in training previous columns is *not* wasted and one can stop *anytime* if the improvement in the multi-class classification performance is negligible. This flexibility also makes our approach dataset dependent to a certain extent [33]. Finally, we note that authors of [38, 39] have discussed choosing $L$ but their work is limited to problems with small $k$.

---

[†] See SI for reason to choose $\lceil \log_2 k \rceil$.

We next perform computational experiments to evaluate the performance of our greedy algorithm and subsequently evaluate the classification performance of the codebooks generated by the greedy algorithm.

## 5 Experiments

We use Gurobi-v9.1 as our integer programming (IP) solver and run all our experiments on a machine with Intel Core i7-6800K CPU, 32GB RAM and 1080Ti Nvidia GPU. We compute optimality gaps of codebooks generated by different approaches using Plotkin's Bound (PB). The gap is defined as: $|\text{PB} - f_{\text{best}}|/f_{\text{best}}$, where $f_{\text{best}}$ denotes the objective function value of the solution. Our code is available at: https://github.com/SamarthGM/Scalable_ecoc.

| $k$ | $L$ | PB | $f_{\text{best}}$ | | | | | Gap (%) | | | | |
|---|---|---|---|---|---|---|---|---|---|---|---|---|
| | | | $\mathcal{IP}1$ | Greedy | [16] | Dense (random) | Dense (best out of 10k) | $\mathcal{IP}1$ | Greedy | [16] | Dense (random) | Dense (best out of 10k) |
| 12 | 24 | 13 | 11 | 12 | 12 | 7 | 9 | 18.18 | **8.33** | **8.33** | 85.7 | 44.4 |
| 16 | 32 | 17 | 14 | 16 | 16 | 9 | 12 | 21.43 | **6.25** | **6.25** | 88.9 | 41.6 |
| 20 | 40 | 21 | 14 | 19 | 19 | 12 | 15 | 50.0 | **10.53** | **10.53** | 75.0 | 40.0 |
| 24 | 48 | 25 | - | 22 | 22 | 15 | 18 | - | **13.64** | **13.64** | 66.7 | 38.8 |
| 28 | 56 | 29 | - | 26 | 25 | 18 | 21 | - | **11.54** | 16.0 | 61.1 | 38.0 |
| 32 | 64 | 33 | - | 29 | 29 | 20 | 24 | - | **13.79** | **13.79** | 65.0 | 37.5 |
| 36 | 72 | 37 | - | 33 | 32 | 23 | 27 | - | **12.12** | 15.62 | 60.9 | 37.0 |
| 40 | 80 | 41 | - | 37 | 36 | 27 | 30 | - | **10.81** | 13.89 | 51.8 | 36.7 |
| 44 | 88 | 45 | - | 40 | 39 | 29 | 33 | - | **12.50** | 15.38 | 55.2 | 24.2 |
| 48 | 96 | 49 | - | 43 | 43 | 33 | 37 | - | **13.95** | **13.95** | 48.5 | 32.4 |

Table 2: Comparison of $\mathcal{IP}1$, Greedy (ours), Gupta and Amin 2021 [16] and Dense codebooks on small problem instances. '-'indicates that no solution is generated within a time limit of 1800 secs.

| $k$ | $L$ | PB | $f_{\text{best}}$ | | | | Gap (%) | | | |
|---|---|---|---|---|---|---|---|---|---|---|
| | | | Greedy | [16] | Dense (random) | Dense (best out of 10k) | Greedy | [16] | Dense (random) | Dense (best out of 10k) |
| 50 | 100 | 51 | 44 | 44 | 34 | 38 | **15.9** | **15.9** | 50.0 | 34.2 |
| 100 | 200 | 101 | 88 | - | 75 | 80 | **14.77** | - | 34.7 | 26.3 |
| 150 | 300 | 151 | 134 | - | 117 | 123 | **12.68** | - | 29.0 | 22.7 |
| 200 | 400 | 201 | 181 | - | 160 | 168 | **11.04** | - | 25.6 | 19.6 |
| 250 | 500 | 251 | 226 | - | 205 | 212 | **11.06** | - | 22.4 | 18.4 |
| 300 | 600 | 301 | 270 | - | 249 | 257 | **11.48** | - | 20.9 | 17.1 |
| 350 | 700 | 351 | 314 | - | 295 | 303 | **11.78** | - | 18.9 | 15.8 |
| 400 | 800 | 401 | 360 | - | 340 | 349 | **11.38** | - | 17.9 | 14.9 |
| 450 | 900 | 451 | 401 | - | 384 | 395 | **12.46** | - | 17.4 | 14.2 |
| 500 | 1000 | 501 | 444 | - | 431 | 441 | **12.83** | - | 16.2 | 13.6 |

Table 3: Comparison of $\mathcal{IP}1$, Greedy (ours), Gupta and Amin 2021 [16] and Dense codebooks on *large* problem instances.

We perform our first set of IP experiments on small problem instances ($k < 50$) to provide a comparison between solving $\mathcal{IP}1$ directly and our Greedy approach (Algorithm 1). Details regarding values of $\rho_1, \rho_2, \gamma$ and their appropriate ranges are provided in SI. We benchmark against the approach of [16] and also against randomly generated Dense codes. The procedure to generate Dense codes [2] is discussed in SI. We summarize our results in Table 2. Notice that for very small instances such as $k = 12$ and $k = 16$, $\mathcal{IP}1$ is able to generate a good quality solution, however for slightly larger problem instances (say $k = 20$) the solution quality of $\mathcal{IP}1$ deteriorates rapidly (the gap is around 50%); and finally for $k \geq 24$, $\mathcal{IP}1$ simply fails to even generate a feasible solution. This highlights the intractability of $\mathcal{IP}1$ to generate good quality codebooks. In contrast, our Greedy approach provides very high quality solutions. Note that for all cases, our greedy approach either matches or outperforms the column-subset selection (CSS) approach of [16]. On very small instances i.e. $k \leq 24$, [16] provides competitive performance to our greedy approach. The reason behind this is because [16] provides a tight IP formulation but at the cost of having exponential $\mathcal{O}(2^k)$ number of binary variables. Their performance deteriorates after $k \geq 24$ as they have to resort to random sampling to mitigate the computational expense of $\mathcal{O}(2^k)$ number of binary variables. It is not surprising that due to this exponential complexity, [16] is not tractable beyond $k > 50$, as discussed next. Also, Dense codes consistently have very high gaps in comparison to our Greedy Approach.

We evaluate the performance of our Greedy approach (Algorithm 1) on large problem instances, i.e. for $k \geq 50$. These results are reported in Table 3. We observe that approach of [16] does not even provide a feasible solution for $k > 50$. In contrast, our greedy approach generates good quality solution for even $k = 500$ while maintaining low optimality gaps. For all cases ($k > 50$) our gaps

are well below 15%. Thus our approach is significantly more tractable than of [16]. Interestingly, for $k \geq 350$ Dense codes also provide competitive performance. To the best of our knowledge, we are the first to achieve such scalability with low optimality-gaps for the codebook design problem.

## 5.1 Classification Performance

We now evaluate and benchmark the classification performance of our codebooks generated using greedy approach on different datasets. For each dataset, we generate codebooks of different sizes using the Greedy approach and compare it with other commonly used codebooks like 1-vs-All, 1-vs-1 [41], Hadamard and Dense codes [2]. For every codebook we train $L$ binary classifiers corresponding to the columns of the codebook. We also benchmark against a multi-class CNN. This multi-class CNN has the same architecture as the binary classifiers except for the size of the output layer, which is 2 for binary classifiers and $k$ for multi-class CNN. For all experiments, we report average of 5 runs. Details regarding different datasets, model architectures, hyper-parameters and training are provided in SI.

**MNIST & CIFAR10:** In our first set of classification experiments, we evaluate the performance on MNIST and CIFAR10 datasets (both with $k = 10$). Using Greedy approach we generate codebooks of size $L = 5$ and 10. The classification performance of different codebooks is reported in tables 4 and 5 respectively. On MNIST, the improvement in accuracy with increase in $L$ is small, since MNIST is a relatively easy classification task. However, on CIFAR10, for which the underlying classification task is much harder than MNIST, we obtain a $1.8\%$ gain in accuracy as the size of the codebook increases from $L = 5$ to $L = 10$. Further on CIFAR10, our *compact* greedy codebooks easily outperforms standard 1-vs-1 codebook which uses $L = 45$ binary classifiers by around $4.5\%$ and also outperforms 1-vs-All codebook by more than $1\%$. For both datasets, Dense and Hadamard codebooks also provides high accuracy, particularly Hadamard codebook on MNIST dataset. In comparison to multi-class CNN, our greedy codebooks provides very similar accuracy on both datasets, with slightly lower accuracy on MNIST but higher on CIFAR10.

| Greedy (Ours) | | | Hadamard | | | 1-vs-All | 1-vs-1 | [16] | Dense | Multi- |
|---|---|---|---|---|---|---|---|---|---|---|
| $L = 5$ | $L = 10$ | $L = 15$ | $L = 5$ | $L = 10$ | $L = 15$ | $L = 10$ | $L = 45$ | $L = 15$ | $L = 15$ | class |
| $97.83 \pm 0.02$ | $98.86 \pm 0.05$ | $\mathbf{98.87 \pm 0.01}$ | $78.89 \pm 0.05$ | $98.67 \pm 0.03$ | $\mathbf{98.89 \pm 0.02}$ | $98.55 \pm 0.07$ | $94.61 \pm 0.12$ | $98.87 \pm 0.01$ | $98.25 \pm 0.04$ | $\mathbf{98.92 \pm 0.04}$ |

Table 4: Accuracy on MNIST dataset.

| Greedy (Ours) | | | Hadamard | | | 1-vs-All | 1-vs-1 | [16] | Dense | Multi- |
|---|---|---|---|---|---|---|---|---|---|---|
| $L = 5$ | $L = 10$ | $L = 15$ | $L = 5$ | $L = 10$ | $L = 15$ | $L = 10$ | $L = 45$ | $L = 15$ | $L = 15$ | class |
| $93.39 \pm 0.08$ | $95.55 \pm 0.06$ | $\mathbf{95.57 \pm 0.02}$ | $74.65 \pm 0.03$ | $95.14 \pm 0.03$ | $95.46 \pm 0.02$ | $94.40 \pm 0.14$ | $90.86 \pm 0.09$ | $95.57 \pm 0.02$ | $95.14 \pm 0.03$ | $95.36 \pm 0.23$ |

Table 5: Accuracy on CIFAR10 dataset.

**CIFAR100 & Caltech-101/256:** In the second set of experiments we evaluate the performance on large class datasets: CIFAR100, Caltech-101 and Caltech-256 with $k = 100, 101$ & $257$ respectively. Recall that, for these large class datasets, the approach of [16] cannot even generate a feasible codebook. Thanks to our greedy algorithm, we can easily generate and evaluate our codebooks on these datasets. Note that the computational expense associated with the training of $L$ binary classifiers for each codebook can be large; however this is a general limitation of ECOC-based classifiers. Firstly, our work addresses this limitation by generating compact codebooks with low-optimality gaps. Secondly, we leverage the power of transfer learning to significantly reduce the training time of binary classifiers. More generally, another motivation to leverage transfer-learning is due to the fact that modern deep-learning models continue to get bigger in size by the day and thus transfer-learning has become even more effective in the face of limited compute and data availability. Our next results indeed demonstrate the additional benefit which ECOCs provide with transfer-learning to further improve classification accuracy.

For transfer-learning we use models trained on ImageNet and replace the last fully-connected layer of 1000 classes with a fully connected DNN with 2 output classes for binary classifiers and $k$ classes for multi-class classifier respectively. We then freeze the weights of all but the last fully-connected DNN. Interestingly, authors in [42] and [45] have recently shown that adversarially trained *robust* models provide better generalization in target domain over *nominally* trained models. Therefore for a more comprehensive evaluation of our approach, we use features from two different ResNet50 models trained on ImageNet. The first model has been pre-trained *nominally* on the ImageNet dataset

---

[†]Authors in [16] report an accuracy of 76.25%, however for a fair comparison we re-train their codebooks using our training procedure, resulting in same accuracy as greedy, since the corresponding $f_{\text{best}}$ value of both codebooks (i.e. greedy and [16]) is same.

while the second model has been trained adversarially with $l_2$-norm perturbations. More details on these individual models are provided in SI. The final multi-class accuracies of all the codebooks and multi-class CNN for both nominal and robust features are provided in tables 6, 7 and 8 respectively.

| Type of feature | Greedy (Ours) | | | Hadamard | | | 1-vs-All | Dense | Multi-class |
|---|---|---|---|---|---|---|---|---|---|
| | $L = 50$ | $L = 100$ | $L = 200$ | $L = 50$ | $L = 100$ | $L = 127$ | $L = 100$ | $L = 200$ | |
| Nominal | $60.62 \pm 0.30$ | $61.65 \pm 0.12$ | $\mathbf{62.15} \pm 0.10$ | $39.32 \pm 0.06$ | $61.98 \pm 0.02$ | $62.04 \pm 0.02$ | $\mathbf{62.58} \pm 0.02$ | $61.70 \pm 0.10$ | $59.19 \pm 0.20$ |
| Robust | $80.85 \pm 0.03$ | $81.46 \pm 0.20$ | $\mathbf{81.68} \pm 0.09$ | $51.82 \pm 0.02$ | $81.51 \pm 0.05$ | $\mathbf{81.7} \pm 0.01$ | $80.41 \pm 0.05$ | $\mathbf{81.60} \pm 0.07$ | $79.59 \pm 0.06$ |

Table 6: Accuracy on CIFAR100 dataset

| Type of feature | Greedy (Ours) | | | Hadamard | | | 1-vs-All | Dense | Multi-class |
|---|---|---|---|---|---|---|---|---|---|
| | $L = 50$ | $L = 100$ | $L = 200$ | $L = 50$ | $L = 100$ | $L = 127$ | $L = 101$ | $L = 200$ | |
| Nominal | $84.55 \pm 0.02$ | $85.04 \pm 0.10$ | $\mathbf{85.33} \pm 0.05$ | $56.49 \pm 0.06$ | $85.10 \pm 0.04$ | $85.17 \pm 0.07$ | $84.06 \pm 0.08$ | $85.16 \pm 0.08$ | $84.73 \pm 0.30$ |
| Robust | $87.08 \pm 0.10$ | $87.52 \pm 0.09$ | $\mathbf{87.76} \pm 0.03$ | $58.06 \pm 0.09$ | $87.55 \pm 0.03$ | $\mathbf{87.73} \pm 0.02$ | $86.57 \pm 0.10$ | $87.50 \pm 0.10$ | $86.39 \pm 0.12$ |

Table 7: Accuracy on Caltech-101 dataset

| Type of feature | Greedy (Ours) | | | Hadamard | | | | 1-vs-All | Dense | Multi-class |
|---|---|---|---|---|---|---|---|---|---|---|
| | $L = 100$ | $L = 200$ | $L = 300$ | $L = 100$ | $L = 200$ | $L = 300$ | $L = 511$ | $L = 257$ | $L = 300$ | |
| Nominal | $76.84 \pm 0.12$ | $77.29 \pm 0.09$ | $\mathbf{77.38} \pm 0.07$ | $32.94 \pm 0.03$ | $75.73 \pm 0.07$ | $\mathbf{77.4} \pm 0.06$ | $\mathbf{77.41} \pm 0.02$ | $75.91 \pm 0.02$ | $77.01 \pm 0.08$ | $76.23 \pm 0.18$ |
| Robust | $77.06 \pm 0.07$ | $77.44 \pm 0.13$ | $\mathbf{77.55} \pm 0.08$ | $33.25 \pm 0.05$ | $75.90 \pm 0.04$ | $77.51 \pm 0.03$ | $\mathbf{77.53} \pm 0.05$ | $76.52 \pm 0.04$ | $\mathbf{77.45} \pm 0.09$ | $76.79 \pm 0.11$ |

Table 8: Accuracy on Caltech-256 dataset

We observe that on all the three (large class) datasets, our greedy codebooks consistently outperform multi-class CNNs. For all cases, robust features indeed provide better accuracy over nominal features thus validating the hypotheses of [42, 45] in the context of ECOC-based classifiers. Importantly, our greedy codebooks provide even further improvement over multi-class CNNs in the robust feature setting.

On *CIFAR100*, in the nominal setting our greedy codebook provides an improvement of around 3% over multi-class CNN and also outperforms other codebooks as well, except 1-vs-All. In the robust setting, both greedy and Hadamard provides high accuracy and an improvement of 2% over multi-class CNN. On *Caltech-101*, greedy codebook provides an improvement of around 0.6% over multi-class CNN in nominal setting and around 1.4% in the robust feature setting. On *Caltech-256* our greedy codebook provides a gain of around 1% over multi-class CNN in both nominal and robust features setting; here the small gain in improvement with robust features over nominal features is consistent with the findings of [42].

It is important to note that Hadamard codes provide high accuracy only for large $L$, thus limiting their application to large multi-class classification problems [5].

These classification results clearly demonstrate the benefit of using ECOC-based classifiers and further highlighting the merits of our element-wise codebook design approach. Importantly, our IP-based Greedy algorithm provides scalability while ensuring small optimality gaps. Further, the resulting codebooks generated, provide improved classification performance on large datasets like CIFAR100, Caltech-101/256. To the best of our knowledge, we are first to report such performance using ECOC-based classifiers.

**Limitation:** The greedy codebooks only provide a marginal improvement over Hadamard and Dense codes in terms of classification accuracy on large-class datasets. However, note that Hadamard codes only provide high accuracy when full Hadamard code of length $L \sim 2^{\lceil \log_2(k) \rceil}$ is used. For much smaller $L$, Hadamard codes perform much worse than our greedy codebooks. Besides, as shown in [16], Dense codes are not robust to adversarial perturbations, but the proposed approach is likely to provide non-trivial adversarial robustness.

## 6 Concluding Remarks, Future Work and Societal Impact

In this paper we developed a scalable approach to solve the discrete codebook design problem for large multi-class problem instances. Our approach generates near-optimal codebooks resulting in higher classification accuracy. As future work, we aim to extend our design approach to other settings, for example, when the cost of miss-classification is different across different class-pairs. Our ECOC based approach is particularly well suited to handle such tasks as we can easily switch from global error-correcting property to class-pair wise error-correcting property [33] with simple modifications. Another extension of our work would be to very large multi-class problems [5] and extreme multi-label (XML) classification problems. We do-not expect any negative societal impact other than that of general classification systems.

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
