# 7  Supplementary Information (SI)

Recall the following definition 1 from the main text:

**Definition 1.** *Hamming distance of a codebook (denoted as $\kappa_{\mathcal{M}}$) is defined as the minimum hamming distance between any two distinct pair of codewords (or rows) in $\mathcal{M}$.*

$$\kappa_{\mathcal{M}} = \min_{(i,j)\in\{1,\ldots k\}^2 | i<j} d_H(\mathcal{M}(i,\cdot), \mathcal{M}(j,\cdot))$$

## 7.1  Proofs:

**Proposition 1** (Error-Correction Capability). *A codebook $\mathcal{M}$ with hamming distance $\kappa_{\mathcal{M}}$, can always correct at-least $\lfloor \frac{\kappa_{\mathcal{M}}-1}{2} \rfloor$ errors.*

*Proof*: Note that every codeword in $\mathcal{M}$ has a distance of at-least $\kappa_{\mathcal{M}}$ from every other codeword in $\mathcal{M}$. This implies that the closed Hamming balls of radius $\lfloor \frac{\kappa_{\mathcal{M}}-1}{2} \rfloor$ are disjoint. Therefore, if a binary vector $q$ differs from some codeword $m \in \mathcal{M}$ in at-most $\lfloor \frac{\kappa_{\mathcal{M}}-1}{2} \rfloor$ places, then $m$ is the unique codeword in $\mathcal{M}$ closest to $q$ . Hence we can safely conclude that the code $\mathcal{M}$ can correct at-least $\lfloor \frac{\kappa_{\mathcal{M}}-1}{2} \rfloor$ errors. ∎

**Lemma 1.** *McCormick relaxation of $\mathcal{IP}1$, denoted as $\mathcal{MC}(\mathcal{IP}1)$, is tight:* $\big\{(z_{ijpq}, x_{ij}, x_{pq}) \in \mathbb{R} \times \{+1,-1\}^2 | z_{ijpq} = x_{ij}x_{pq}\big\} \equiv \big\{(z_{ijpq}, x_{ij}, x_{pq}) \in \mathbb{R} \times \{+1,-1\}^2 | (2a), (2b)\big\}.$

$$\text{Lower:} \qquad z_{ijpq} \geq -x_{ij} - x_{pq} - 1; \ z_{ijpq} \geq \ x_{ij} + x_{pq} - 1 \qquad (2a)$$

$$\text{Upper:} \qquad z_{ijpq} \leq \ x_{ij} - x_{pq} + 1; \ z_{ijpq} \leq -x_{ij} + x_{pq} + 1 \qquad (2b)$$

*Proof*: Since $x_{ij}$ and $x_{pq}$ are integer variables, therefore the McCormick inequalities in (2) ensure that $z_{ijpq}$ also takes integer values i.e. $z_{ijpq} \in \{+1,-1\}$, thus $z_{ijpq} = x_{ij}x_{pq}$ and (2) are equivalent. This can also be verified by simple enumeration as shown below:

| $x \in \{-1, +1\}$
$y \in \{-1, +1\}$
$z \geq -x - y - 1$
$z \geq \ x + y - 1$
$z \leq -x + y + 1$
$z \leq \ x - y + 1$ | Case I:
$x = -1; y = -1$
$\left.\begin{array}{l} z \geq \ 1 \\ z \geq -3 \\ z \leq \ 1 \\ z \leq \ 1 \end{array}\right\} \Rightarrow z = 1$ | Case II:
$x = -1; y = 1$
$\left.\begin{array}{l} z \geq -1 \\ z \geq -1 \\ z \leq \ 3 \\ z \leq -1 \end{array}\right\} \Rightarrow z = -1$ | Case III:
$x = 1; y = -1$
$\left.\begin{array}{l} z \geq -1 \\ z \geq -1 \\ z \leq -1 \\ z \leq \ 3 \end{array}\right\} \Rightarrow z = -1$ |
| --- | --- | --- | --- |
| Case IV:
$x = 1; y = 1$
$\left.\begin{array}{l} z \geq -3 \\ z \geq \ 1 \\ z \leq \ 1 \\ z \leq \ 1 \end{array}\right\} \Rightarrow z = 1$ | | | |

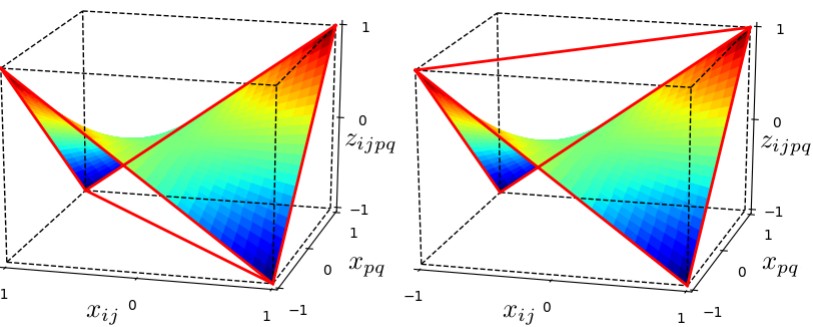

Figure 4: Lower and Upper McCormick envelopes for bilinear term $z_{ijpq} = x_{ij}x_{pq}$. (Image adapted from [6]) ∎

**Lemma 2.** *Let $\mathcal{M}$ and $\tilde{\mathcal{M}}$ be two binary error-correcting code of size $k \times l$ and $k \times (l+\tilde{l})$ respectively, where $\tilde{l} \geq 0$, such that all columns in $\mathcal{M}$ are also in $\tilde{\mathcal{M}}$, i.e. $\mathcal{M} \subseteq \tilde{\mathcal{M}}$. Then, the Hamming distances (recall definition 1) of $\mathcal{M}$ and $\tilde{\mathcal{M}}$ satisfy: $\kappa_{\mathcal{M}} \leq \kappa_{\tilde{\mathcal{M}}}$, implying that the error-correcting capability of $\tilde{\mathcal{M}}$ is atleast as good as $\mathcal{M}$.*

*Proof*: Consider a codebook $\mathcal{M}$ of size $k \times l$ which has a hamming distance of $\kappa_{\mathcal{M}}$, where $\kappa_{\mathcal{M}}$ is defined as :

$$\kappa_{\mathcal{M}} = \min_{(i,j)\in\{1,\ldots k\}^2 | i<j} d_H(\mathcal{M}(i,\cdot), \mathcal{M}(j,\cdot))$$

Now another codebook $\tilde{\mathcal{M}}$ of size $k \times (l + \tilde{l})$ is obtained by adding $\tilde{l}$ columns to $\mathcal{M}$. Since Hamming distance of any $(i,j)$ pair of rows in $\tilde{\mathcal{M}}$ can be either greater or equal to the Hamming distance of corresponding $(i,j)$ pair of rows in $\mathcal{M}$, due the monotonic nature of Hamming distance, we can write:

$$d_H(\mathcal{M}(i,\cdot), \mathcal{M}(j,\cdot)) \leq d_H(\tilde{\mathcal{M}}(i,\cdot), \tilde{\mathcal{M}}(j,\cdot)) \qquad \forall (i,j) \in \{1, \ldots k\}^2 | i < j$$

Since the above inequality holds for every pair of rows, therefore:

$$\underbrace{\min_{(i,j)\in\{1,\ldots k\}^2 | i<j} d_H(\mathcal{M}(i,\cdot), \mathcal{M}(j,\cdot))}_{= \kappa_{\mathcal{M}}} \leq \underbrace{\min_{(i,j)\in\{1,\ldots k\}^2 | i<j} d_H(\tilde{\mathcal{M}}(i,\cdot), \tilde{\mathcal{M}}(j,\cdot))}_{= \kappa_{\tilde{\mathcal{M}}}}$$

$$\kappa_{\mathcal{M}} \leq \kappa_{\tilde{\mathcal{M}}}$$

$\blacksquare$

**Proposition 2.** *For a codebook $\mathcal{M}$, if there exists an assignment of codeword $c_i \in \mathcal{C}(\tilde{l})$ to each vertex of graph $\mathcal{G}_{\mathcal{M}}$ such that no two connected vertices receive the same codeword, then the hamming distance of the codebook $\kappa_{\mathcal{M}}$ can be increased by atleast 1 by adding codewords from $\mathcal{C}(\tilde{l})$ to the rows of $\mathcal{M}$.*

*Proof*: Recall the following two important observations:

1. Increasing the hamming distance of row-pairs in $\mathcal{E}_{\mathcal{M}}$ by atleast 1 through addition of columns (or equivalently by appending codewords $c_i$) to $\mathcal{M}$, effectively increases the minimum hamming distance of the entire codebook by atleast 1.

2. Any pair of distinct codewords in the set $\mathcal{C}(\tilde{l})$ differ by at least 1 and atmost $\tilde{l}$, i.e. :
   $1 \leq d_H(c_i, c_j) \leq \tilde{l} \ \forall c_i, c_j \in \mathcal{C}(\tilde{l}), \ i \neq j$

If we can assign codeword $c_i$ from $\mathcal{C}(\tilde{l})$ to every row in $\mathcal{M}$ such that for every row-pair $(i,j)$ in $\mathcal{E}_{\mathcal{M}}$ satisfies $d_H(c_i, c_j) \geq 1$, then because of the first observation, it is ensured that the minimum hamming distance of the entire codebook $\mathcal{M}$ increases by at-least 1.

Finally, to ensure that $d_H(c_i, c_j) \geq 1$ for all row-pairs in $\mathcal{E}_{\mathcal{M}}$, it is sufficient to ensure that $c_i \neq c_j$ for all $(i,j)$ row-pairs in $\mathcal{E}_{\mathcal{M}}$ because of the second observation. This is same as ensuring that the vertices of $\mathcal{G}_{\mathcal{M}}$ are assigned different codewords (colors), since $(i,j)$ pairs in $\mathcal{E}_{\mathcal{M}}$ forms the edges of $\mathcal{G}_{\mathcal{M}}$. $\blacksquare$

**Theorem 1.** *For any binary code $\tilde{\mathcal{M}}$ resulting from adding $\tilde{l}$ columns to an existing binary code $\mathcal{M}$, the following holds regarding the hamming distance (recall definition 1) $\kappa_{\tilde{\mathcal{M}}}$ of the code $\tilde{\mathcal{M}}$:*

- *If the chromatic number of the graph $\mathcal{G}_{\mathcal{M}}$, $\xi(\mathcal{G}_{\mathcal{M}})$ is greater than size of the set of possible codewords $\mathcal{C}(\tilde{l})$, i.e. $\xi(\mathcal{G}_{\mathcal{M}}) > |\mathcal{C}(\tilde{l})|$, then $\kappa_{\tilde{\mathcal{M}}} = \kappa_{\mathcal{M}}$.*

- *In particular, for $\tilde{l} \in \{1, 2\}$ :*

$$\kappa_{\tilde{\mathcal{M}}} \leq \begin{cases} \kappa_{\mathcal{M}} + \tilde{l}, & \text{if } \xi(\mathcal{G}_{\mathcal{M}}) = 2 \\ \kappa_{\mathcal{M}} + \tilde{l} - 1, & \text{if } 3 \leq \xi(\mathcal{G}_{\mathcal{M}}) \leq 4 \\ \kappa_{\mathcal{M}}. & \text{if } \xi(\mathcal{G}_{\mathcal{M}}) \geq 5 \end{cases}$$

*Proof*: In proposition 2, we have already established the connection between vertex-coloring of $\mathcal{G}_{\mathcal{M}}$ and the increase in Hamming distance of $\mathcal{M}$, i.e. $\kappa_{\mathcal{M}}$. The first part of the theorem automatically

follows that since in total we have $|\mathcal{C}(\tilde{l})|$ number of codewords (or colors) at our disposal, but if the minimum number of colors required to color $\mathcal{G}_\mathcal{M}$ is strictly greater than $|\mathcal{C}(\tilde{l})|$, i.e. $\xi(\mathcal{G}_\mathcal{M}) > |\mathcal{C}(\tilde{l})|$, implies that for at-least one $(i, j)$ row-pair in $\mathcal{E}_\mathcal{M}$, $d_H(c_i, c_j) = 0$ as $c_i = c_j$. Thus the Hamming distance of the resulting codebook will not increase, or $\kappa_{\tilde{\mathcal{M}}} = \kappa_\mathcal{M}$.

To prove second part of the theorem, we only need to take care of the cases when $\xi(\mathcal{G}_\mathcal{M}) \leq |\mathcal{C}(\tilde{l})|$.

- Case I: $\xi(\mathcal{G}_\mathcal{M}) = 2$, implies that we need only two colors or codewords.
  For $\tilde{l} = 1$, $\mathcal{C}(1) = \{[-1], [+1]\}$, using the only possible combination i.e. $\{[-1], [+1]\}$, the hamming distance can be increased by $1$.
  For $\tilde{l} = 2$, $\mathcal{C}(2) = \{[-1, -1], [+1, +1], [-1, +1], [+1, -1]\}$, as we need to pick two codewords from $\mathcal{C}(2)$, therefore there are $\binom{4}{2} = 6$ possible valid combinations. However out of these $6$ combinations, a *maximal* gain of $2$ can only be achieved by using either $\{[-1, -1], [+1, +1]\}$ or $\{[-1, +1], [+1, -1]\}$. The remaining $4$ combinations will only lead to a gain of $1$.
- Case II: $\xi(\mathcal{G}_\mathcal{M}) = 3$, implies that we need $3$ distinct codewords.
  For $\tilde{l} = 1$, we need to pick $3$ codewords from $\mathcal{C}(1) = \{[-1], [+1]\}$, but since there are only two codewords, therefore the hamming distance cannot be increased.
  For $\tilde{l} = 2$, $\mathcal{C}(2) = \{[-1, -1], [+1, +1], [-1, +1], [+1, -1]\}$, we need to pick $3$ codewords from $\mathcal{C}(2)$, therefore there are $\binom{4}{3} = 4$ possible valid combinations. In all the four different possible combinations there will be codewords with hamming distance of $1$, therefore the maximal gain of only $1$ can be achieved.
- Case III: $\xi(\mathcal{G}_\mathcal{M}) = 4$, implies that we need $4$ distinct codewords, similar to previous case for $\tilde{l} = 1$ there are only two codewords therefore the hamming distance cannot be increased.
  For $\tilde{l} = 2$, we need to pick $4$ codewords from $\mathcal{C}(2)$. Since $|\mathcal{C}(2)| = 4$, therefore we need to all $4$ codewords in $\mathcal{C}(2)$. Here again there will be codewords with hamming distance of $1$, therefore the maximal gain of only $1$ can be achieved. ∎

**Generalization of Theorem 1 to cases when $\tilde{l} \geq 3$:**

Following our previous discussion it is not hard to see that we can trivially upper and lower bound $\kappa_{\tilde{\mathcal{M}}}$ as follows:

$$\kappa_\mathcal{M} \leq \kappa_{\tilde{\mathcal{M}}} \leq \kappa_\mathcal{M} + \tilde{l} \tag{9}$$

From (9), we know that $\kappa_{\tilde{\mathcal{M}}} \in \left\{\kappa_\mathcal{M}, \ \kappa_\mathcal{M} + 1, \ \ldots, \ \kappa_\mathcal{M} + \tilde{l}\right\}$, as $\kappa_{\tilde{\mathcal{M}}}$ can only take positive integer values.

From the first part of theorem 1, we already know that if the chromatic-number of the graph $\mathcal{G}_\mathcal{M}$ is strictly greater than the size of the set of codewords $\mathcal{C}(\tilde{l})$, i.e. $\xi(\mathcal{G}_\mathcal{M}) > |\mathcal{C}(\tilde{l})|$, then $\kappa_{\tilde{\mathcal{M}}} = \kappa_\mathcal{M}$.

Also, if $\xi(\mathcal{G}_\mathcal{M}) = |\mathcal{C}(\tilde{l})|$, then we need to pick $|\mathcal{C}(\tilde{l})|$ number of codewords from $\mathcal{C}(\tilde{l})$, i.e. we need to pick all the codewords in $\mathcal{C}(\tilde{l})$, we can only get a maximal gain of $1$, therefore we have $\kappa_{\tilde{\mathcal{M}}} \leq \kappa_\mathcal{M} + 1$.

Further, if $\xi(\mathcal{G}_\mathcal{M}) = 2$, then we need to pick $2$ codewords from $\mathcal{C}(\tilde{l})$, and by choosing complementary codewords such as $\{\underbrace{[1, \ldots, 1]}_{\tilde{l} \text{ entries}}, \underbrace{[-1, \ldots, -1]}_{\tilde{l} \text{ entries}}\}$, we can get a maximal gain of $\tilde{l}$, thus we have:

$\kappa_{\tilde{\mathcal{M}}} \leq \kappa_\mathcal{M} + \tilde{l}$.

Therefore we have taken care of the cases when $\xi(\mathcal{G}_\mathcal{M}) = 2$ and $\xi(\mathcal{G}_\mathcal{M}) \geq |\mathcal{C}(\tilde{l})|$. Now since $\xi(\mathcal{G}_\mathcal{M})$ can only take integer values, therefore the remaining cases are $\xi(\mathcal{G}_\mathcal{M}) \in \{3, \ldots, |\mathcal{C}(\tilde{l})| - 1\}$. Now solving for the remaining cases by enumeration can be quite tedious and prone to human error, therefore we provide an automated optimization based solution approach.

Recall that we have already discussed earlier that $\kappa_{\tilde{\mathcal{M}}} \in \left\{\kappa_\mathcal{M}, \ \kappa_\mathcal{M} + 1, \ \ldots, \ \kappa_\mathcal{M} + \tilde{l}\right\}$. Consider the following question: What is the maximum number of codewords that can be selected from $\mathcal{C}(\tilde{l})$

such that each of these selected codewords are atleast at a Hamming-distance of $\tilde{d}$ apart from each other? This can be easily answered by solving a integer program which we discuss next. We assign a binary decision variable $x_i \in \{0, 1\} \ \forall \ i \in \{1, \ldots, |\mathcal{C}(\tilde{l})|\}$ to each codeword $c_i$ in $\mathcal{C}(\tilde{l})$ representing the outcome that whether or not the codeword $c_i$ is selected. Further to ensure that codewords only with Hamming-distance of atleast $\tilde{d}$ apart are selected, we add a constraint $x_i + x_j \leq 1$ for every pair of codewords in $\mathcal{C}(\tilde{l})$ for which the Hamming distance is strictly below $\tilde{d}$. Finally, we set the objective function to $\sum_i x_i$. The IP is given as:

$$\max_x \ \sum_i x_i \tag{10a}$$

$$\text{s.t.} \tag{10b}$$

$$x_i + x_j \leq 1 \quad \forall \ (i,j) \ \in \ \left\{\{1, \ldots, |\mathcal{C}(\tilde{l})|\} \times \{1, \ldots, |\mathcal{C}(\tilde{l})|\} | d_H(c_i, c_j) < \tilde{d}\right\} \tag{10c}$$

$$x_i \ \in \ \{0,1\} \quad \forall \ i \ \in \{1, \ldots, |\mathcal{C}(\tilde{l})|\} \tag{10d}$$

For a given $\tilde{l}$ and its corresponding set of codewords $\mathcal{C}(\tilde{l})$, we can easily solve the above IP for different values of $\tilde{d} \in \{1, \ldots, \tilde{l}\}$. From the resulting objective function values, we can complete the earlier remaining cases when $\xi(\mathcal{G}_{\mathcal{M}}) \in \{3, \ldots, |\mathcal{C}(\tilde{l})| - 1\}$. To avoid introduction of new notation and complexity, we explain on how to complete for the remaining cases, using simple examples.

For $\tilde{l} = 3$, we solve the above IP with $\tilde{d} \in \{1, 2, 3\}$, and get $8, 4, 2$ as objective function values for each $\tilde{d}$ respectively. This means that from the set $\mathcal{C}(3)$, we can select atmost $8$ codewords at a distance $1$, $4$ codewords at distance $2$ and $2$ codewords at a distance $3$. Using this information we can easily infer the following:

$$\kappa_{\tilde{\mathcal{M}}} \leq \begin{cases} \kappa_{\mathcal{M}} + 3, & \text{if } \xi(\mathcal{G}_{\mathcal{M}}) = 2 \\ \kappa_{\mathcal{M}} + 2, & \text{if } 3 \leq \xi(\mathcal{G}_{\mathcal{M}}) \leq 4 \\ \kappa_{\mathcal{M}} + 1, & \text{if } 5 \leq \xi(\mathcal{G}_{\mathcal{M}}) \leq 8 \\ \kappa_{\mathcal{M}}. & \text{if } \xi(\mathcal{G}_{\mathcal{M}}) \geq 9 \end{cases}$$

For $\tilde{l} = 4$, we solve the above IP with $\tilde{d} \in \{1, 2, 3, 4\}$, and get $16, 8, 2, 2$ as objective function values for each $\tilde{d}$ respectively. We can easily infer the following:

$$\kappa_{\tilde{\mathcal{M}}} \leq \begin{cases} \kappa_{\mathcal{M}} + 4, & \text{if } \xi(\mathcal{G}_{\mathcal{M}}) = 2 \\ \kappa_{\mathcal{M}} + 2, & \text{if } 3 \leq \xi(\mathcal{G}_{\mathcal{M}}) \leq 8 \\ \kappa_{\mathcal{M}} + 1, & \text{if } 9 \leq \xi(\mathcal{G}_{\mathcal{M}}) \leq 16 \\ \kappa_{\mathcal{M}}. & \text{if } \xi(\mathcal{G}_{\mathcal{M}}) \geq 17 \end{cases}$$

Using the procedure demonstrated above for $\tilde{l} = 3$ and $\tilde{l} = 4$, upper bounds on $\kappa_{\tilde{\mathcal{M}}}$ can be generated easily for any $\tilde{l}$.

## 7.2 Equivalent reformulation of $\mathcal{IP}1$ using $L_1$-norm :

One may be lead to think that the inherent difficulty in solving $\mathcal{IP}1$ comes from bilinear constraints (1b)-(1c). We now present an alternative formulation of the optimal element-wise codebook design problem. This formulation, denoted $\mathcal{IP}3$ is a mixed integer linear program (MILP) instead of the MIQCP $\mathcal{IP}1$. This alternative formulation leverages a useful property of binary vectors: for any $p, q \in \{+1, -1\}^{r \times 1}$, the Hamming distance (i.e. $l_0$-norm), $l_1$-norm and $l_2$-norm are related as follows:

$$\underbrace{\sum_{i=1}^r \mathbb{1}_{\{p_i \neq q_i\}}}_{l_0\text{-norm}} = \underbrace{\frac{1}{2} \sum_{i=1}^r |p_i - q_i|}_{l_1\text{-norm}} = \underbrace{\frac{1}{4} \sum_{i=1}^r (p_i - q_i)^2}_{l_2^2\text{-norm}}$$

Thus, by modeling Hamming distances using $l_1$-norm, we can reformulate our codebook design problem as the following integer program:

$$\mathcal{IP}3 : \max_{x} \ \min \ \{d_H^{1,2}(x), \ d_H^{1,3}(x), \ \ldots, d_H^{k-1,k}(x)\}$$

$$d_H^{i,\hat{i}}(x) = \frac{1}{2} \sum_{j=1}^{L} |x_{ij} - x_{\hat{i}j}| \qquad \forall \, (i,\hat{i}) \in \mathcal{P}_{\mathcal{N}} \tag{11a}$$

$$\rho_1 \le \frac{1}{2} \sum_{i=1}^{k} |x_{ij} - x_{i\hat{j}}| \le \rho_2 \qquad \forall \, (j,\hat{j}) \in \mathcal{P}_{\mathcal{T}} \tag{11b}$$

$$-\gamma \le \sum_{i=1}^{k} n_i x_{ij} \le \gamma \qquad \forall \, j \in \mathcal{T} \tag{11c}$$

$$x_{ij} \in \{+1, -1\} \qquad \forall \, (i,j) \in \mathcal{N} \times \mathcal{T} \tag{11d}$$

The absolute value operator $|\cdot|$ in $\mathcal{IP}3$ can be simplified using big-M constraints. Denoting $y' = |y|$, we can lower and upper bound $y'$ as follows:

$$\text{Lower:} \quad y' \ge y \qquad ; y' \ge -y \tag{12}$$

$$\text{Upper:} \quad y' \le y + Mz \quad ; y' \le -y + M(1-z) \tag{13}$$

where $M$ is chosen such that $M \ge 2|y|$ and $z$ is an auxiliary binary variable ($z \in \{0,1\}$). For our case, choosing $M = 4$ is sufficient to ensure that (12) and (13) leads to desired linearization of (11a) and (11b)) in $\mathcal{IP}2$. However, this linearization comes at the cost of introducing additional *binary* variables and constraints. Similar to $\mathcal{IP}1$, we here have $\mathcal{O}(k^3)$ abs-value operators, and therefore we will have $\mathcal{O}(k^3)$ additional binary variables. As we already have $k \times L \approx k^2$ binary variables corresponding to the entries of our codebook, in total we will have $\mathcal{O}(k^3)$ number of binary variables. In summary, although $\mathcal{IP}3$ is a linear formulation, it involves $\mathcal{O}(k^3)$ binary variables in contrast to $\mathcal{O}(k^2)$ in the McCormick relaxation i.e. $\mathcal{MC}(\mathcal{IP}1)$ of $\mathcal{IP}1$.

In practice we observe that $\mathcal{IP}3$ suffers from the same computational challenge as $\mathcal{IP}1$, i.e. its LP-relaxation is quite loose for tractably solving practical instances of the codebook design problem.

### 7.3   Choosing values of $\rho_1, \rho_2$ & $\gamma$:

**Choosing $\rho_1$ and $\rho_2$:**   In our experiments we observed that the final accuracy is not highly sensitive to as long as $\rho_1$ and $\rho_2$ are defined in a reasonable range. To avoid exactly same columns $\rho_1 \ge 1$ and to avoid complementary columns $\rho_2 \le k - 1$. We used $\rho_1 = \lfloor k/3 \rfloor$ and $\rho_2 = \lfloor 2k/3 \rfloor$.

**Choosing $\gamma$:**   To better understand how $\gamma$ enforces the balanced column critera, lets consider a dataset in which each class has same number of data-points i.e. $n_i = n_j \ \forall \ (i,j) \in \{1, \ldots, k\}$, where $i \ne j$ and without loss of generality $k$ is even. In this setting, the valid range of $\gamma$ is given as $0 \le \gamma \le k - 2$. If $\gamma = 0$, then a hard balanced column criteria will be enforced, i.e. both the binary classes will have exactly the same number of data-points and if $\gamma = k - 2$, then maximum amount of imbalance is allowed.

A smaller value of $\gamma$ makes $\mathcal{IP}2$ tighter, consequently faster to solve and overall improves the error-correcting capability of the final codebook. However, in such cases, as the resulting columns are highly balanced, therefore training binary classifiers become relatively hard, in particular for simple linear models [2, 9, 41] and thus overall adversely affecting final classification accuracy. On the contrary, imbalanced columns are easier to learn for simple (linear) models however can be challenging for complex models like DNNs (requiring to choose the training loss function carefully). Therefore while choosing the value of $\gamma$, the complexity of the binary classifier should also be considered.

In in our experiments when $k$ is small, i.e $k = 10$ for MNIST and CIFAR10, we observed that the final classification accuracy was not highly sensitive to the value of $\gamma$. On large class experiments i.e. $k \ge 50$, we observed that in a very hard balanced column criteria setting, where we set $\gamma$ to low values such that only near 50%-50% splits are allowed, in this setting we actually observed a slight reduction in the final accuracy. Therefore, we used an intermediary value of $\gamma$ such that splits of at most 60%-40% or 65%-35% are allowed.

## 7.4 Decoding scheme:

We use a class score based decoding scheme same as [16]. We compute a score for each class by carefully adding the output score of each binary classifier logit depending upon whether the entry of that class (or row) has +1 or -1. Normalization is not required for binary codes as each row of a binary code has the same number of entries. 1-vs-1 (a ternary code) is balanced in the sense that each row has the same number of zero entries hence normalization is not required. In fact we observed that score based decoding consistently provides slightly better accuracy over Hamming decoding. Further Hamming decoding has issues as pointed out in [39].

## 7.5 Integer Programming Formulation to solve vertex-coloring

For the graph $\mathcal{G}_{\mathcal{M}}$ with set of vertices as $\mathcal{V}$ and set of edges as $\mathcal{E}_{\mathcal{M}}$, the vertex-coloring problem can be solved by solving the following integer program:

$$\min_{w} \sum_{i=1}^{H} w_i \tag{14a}$$

$$\text{s.t.}$$

$$\sum_{i=1}^{H} x_{vi} = 1 \qquad \forall\, v \in \mathcal{V} \tag{14b}$$

$$x_{ui} + x_{vi} \leq w_i \qquad \forall\, (u,v) \in \mathcal{E}_{\mathcal{M}}, i = 1, \ldots, H \tag{14c}$$

$$x_{vi} \in \{0,1\} \qquad \forall\, v \in \mathcal{V}, i = 1, \ldots, H$$

$$w_i \in \{0,1\} \qquad \forall\, i = 1, \ldots, H$$

In the above IP, $w_i$'s are the binary decision variables representing the outcome whether or not the $i$-th color is used to color any vertex; $x_{vi}$'s represent the binary decision whether or not, the vertex $v$ is colored using $i$-th color. (14a) computes the chromatic number of the graph; constraint (14b) ensures that each vertex is assigned only a single color; constraint (14c) ensures that the vertices of the edge $(u,v)$ are assigned different colors.

Note that $H$ is a pre-computed upper-bound on the chromatic number. A simple upper-bound on chromatic number is $\max(\deg(\mathcal{G}_{\mathcal{M}})) + 1$. Therefore, we can solve the above IP with $H = \max(\deg(\mathcal{G}_{\mathcal{M}})) + 1$.

### 7.5.1 Using vertex-coloring solution to generate a feasible solution to $\mathcal{IP}2$

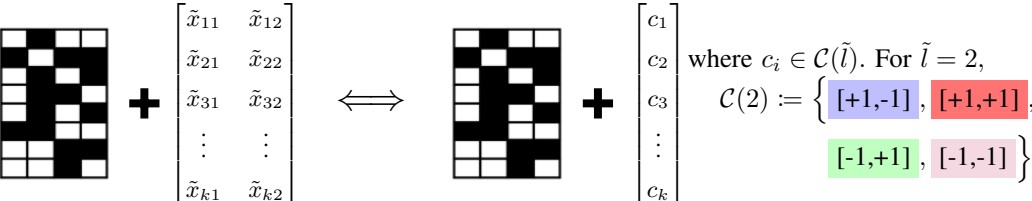

Figure 5: Addition of columns is equivalent to addition of codewords.

Recall that the main motivation behind theorem 1, was that addition of $\tilde{l}$ number of columns to an existing code can also be viewed as addition of codewords $c_i$ to each row of the codebook, where $c_i \in \mathcal{C}(\tilde{l})$; see figure 5. Thanks to theorem 1, using the optimal objective function value obtained after solving the vertex-coloring IP, we can obtain an upper bound to $\mathcal{IP}2$. But note that we also obtain a valid color assignment to each vertex in the graph $\mathcal{G}_{\mathcal{M}}$ such that no two connected vertices have the same color. As shown in figure 5 (on the left), to every color, we can assign a codeword and thus also providing us with the values for the entries $\tilde{x}_{ij}$ in the two columns, therefore providing us with a *possibly feasible* solution to $\mathcal{IP}2$. However this solution may not always be a feasible solution to $\mathcal{IP}2$, depending on the hardness of the column separation and balanced column criteria

constraints. Interestingly, we can refactor this infeasible solution to generate a feasible solution using the procedure discussed next.

We first make an important observation regarding the structure of $\mathcal{G}_\mathcal{M} = (\mathcal{V}, \mathcal{E}_\mathcal{M})$. In practice, we observe that a large number of vertices in $\mathcal{V}$ have degree 0. We denote the set of zero-degree vertices with $\mathcal{V}_0$ and the remaining set of vertices as $\mathcal{V}_h$.

This also means that any row in $\mathcal{M}$, corresponding to a vertex in $\mathcal{V}_0$, already has a hamming distance of at-least $\kappa_\mathcal{M} + 1$ to all other remaining $k - 1$ rows in $\mathcal{M}$. Further note that vertices in $\mathcal{V}_0$ can be assigned any color out of $\xi(\mathcal{G}_\mathcal{M})$ number of colors, without violating vertex-coloring.

Since it doesn't matter which color is assigned to vertices in $\mathcal{V}_0$, therefore it also doesn't matter which codeword is assigned to the rows corresponding to vertices in $\mathcal{V}_0$. We exploit this to refactor the above discussed infeasible solution into a feasible solution to $\mathcal{IP}2$. Instead of assigning codewords to all $k$ rows, we assign codewords to only those rows corresponding to vertices in $\mathcal{V}_h$, thus in a way assigning value to $\tilde{x}_{ij} \ \ \forall \ i \ \in \mathcal{V}_h, \ \ j \in \tilde{\mathcal{T}}$. We therefore do-not assign any value to $\tilde{x}_{ij} \ \ \forall \ i \ \in \mathcal{V}_0, \ \ j \in \tilde{\mathcal{T}}$. We provide this *partial* solution to Gurobi as a possible initial solution with the optimization model as $\mathcal{IP}2$. A useful feature of Gurobi solver is that if a partial solution is provided by the user, then instead of outrightly rejecting this partial solution, Gurobi first tries to complete this solution by trying to fill the missing values such that all the model constraints are satisfied. If the partial solution results in a feasible solution then Gurobi may start branch-and-bound (B&B) procedure if this feasible solution is not optimal. In practice, we observe that in most cases the feasible solution is actually optimal and therefore Gurobi does not even have to start the B&B procedure. However if the partial solution does not result in a feasible solution then Gurobi starts (B&B) procedure as usual.

Finally, one thing to note is that different codewords can be assigned to different colors thus resulting in different initial *partial* solutions. Few permutation for $\tilde{l} = 2$ and $\xi(\mathcal{G}_\mathcal{M}) = 4$ are shown in figure 6. Therefore if the partial solution from any particular permutation fails to result in a feasible, a different permutation can be tried.

$$\mathcal{C}(2) := \Big\{ \ [+1,-1] \ , \ [+1,+1] \ , \ [-1,+1] \ , \ [-1,-1] \ \Big\}$$

$$\mathcal{C}(2) := \Big\{ \ [+1,-1] \ , \ [+1,+1] \ , \ [-1,+1] \ , \ [-1,-1] \ \Big\}$$

$$\mathcal{C}(2) := \Big\{ \ [+1,-1] \ , \ [+1,+1] \ , \ [-1,+1] \ , \ [-1,-1] \ \Big\}$$

$$\mathcal{C}(2) := \Big\{ \ [+1,-1] \ , \ [+1,+1] \ , \ [-1,+1] \ , \ [-1,-1] \ \Big\}$$

Figure 6: Some different permutations of codewords assigned to individual colors.

## 7.6 Details of different datasets:

| Dataset | Number of classes $k$ | Number of Training samples | Number of Test samples |
|---|---|---|---|
| MNIST [27] | 10 | 60,000 | 10,000 |
| CIFAR10 [25] | 10 | 50,000 | 10,000 |
| CIFAR100 [25] | 100 | 50,000 | 10,000 |
| Caltech-101 [10] | 101 | 3,030 | 5,647 |
| Caltech-256 [15] | 257 | 15,420 | 15,187 |

### 7.7 Details regarding the network architecture, hyperparameters and training for different experiments:

#### 7.7.1 MNIST/CIFAR10:

**MNIST:**
For all our experiments on the MNIST dataset we use the following network architecture, where the number of output classes is 10 for a multi-class classifier and 2 for any binary classifier.

```
model  =nn.Sequential(
        nn.Conv2d(1, 32, 5, stride=1, padding=2),
        nn.ReLU(),
        nn.MaxPool2d((2, 2), stride=(2, 2), padding=0),
        nn.Conv2d(32, 64, 5, stride=1, padding=2),
        nn.ReLU(),
        nn.MaxPool2d((2, 2), stride=(2, 2), padding=0),
        Flatten(),
        nn.Linear(64 × 7 × 7, 1024),
        nn.ReLU(),
        nn.Linear(1024, n_classes))
```

In the above model, Flatten() layer is defined using the following class:
```
class Flatten(nn.Module):
    def forward(self, input):
         return input.view(input.size(0), -1)
```

We use a learning rate lr = 0.01, batch size of 128 and train for a total of 50 epochs and reduce the learning rate by a factor of 10 after 25 epochs. We use cross-entropy as our classification loss function.

**CIFAR10:** For all our experiments on CIFAR10 dataset, we use a ResNet-18 network architecture [19] where the size of output FC-layer is 10 for multi-class classifier and 2 for binary classifiers. We use a batch size of 128, with an initial learning rate lr=0.1, momentum=0.9 and weight decay $=5e-4$. We train for 200 epochs and reduce the learning rate by a factor of 5 after 60,120 and 160 epochs. We use the following standard data-augmentation methods during training:

```
TRAIN_TRANSFORMS=transforms.Compose([
transforms.RandomCrop(32, padding=4),
transforms.RandomHorizontalFlip(),
transforms.ToTensor(),
normalize])

TEST_TRANSFORMS=transforms.Compose([
transforms.ToTensor(),
normalize])
```

#### 7.7.2 CIFAR100, Caltech-101, Caltech-256 (Transfer Learning):

We now discuss the details of our transfer-learning experiments on CIFAR100, Caltech-101, Caltech-256. For each of these three datasets we leverage fixed feature transfer-learning. Further, for each dataset we experiment with two different types of models trained on ImageNet provided by [42]. The first model is nominally trained while the second model is adversarially trained.

*Nominally* trained model: A ResNet-50 model trained on ImageNet using SGD with batch size of 512, momentum of 0.9, and weight decay of $1e-4$. It is trained for 90 epochs with an initial learning rate of 0.1 that drops by a factor of 10 every 30 epochs. We use the same pre-trained model across all the three datasets. The standard natural accuracy of $75.8\%$ is achieved.

Adversarially trained *Robust* model: A ResNet-50 model trained on ImageNet using adversarial training [31]. It is trained on adversarial examples generated within maximum allowed $l_2$-norm based $\epsilon$ perturbations using 3 attack steps and a step size of $\frac{2}{3}\epsilon$.

We select $\epsilon = \{3, 1, 0.05\}$ for CIFAR100, Caltech-101 and Caltech-256 respectively [42]. Standard natural accuracy of $62.83\%, 70.43\%$ and $75.59\%$ is achieved for each of the individual models respectively.

**Training multi-class classifier:**
For training multi-class classifier, we freeze the weights of all the layers except the last fully-connected layer. Therefore the final feature layer is of size 2048 as we use ResNet-50 models. We next replace the last fully-connected (FC) output layer of size 1000 with a small fully connected deep neural network with 3 hidden layers of size $2000 - 1000 - 500$ and the final output layer of size $k$. The weights of the small DNN are randomly initialized. We use multi-class cross-entropy classification loss. We train only this fully-connected layer for 150 epochs using SGD with batch size of 64, momentum of 0.9, weight decay of $5e - 4$. We tested with different initial learning rate lr $\in \{0.1, 0.01, 0.001\}$ and the learning rate drops by a factor of 10 after every 50 epochs. For all the datasets, initial lr $= 0.1$ yielded best results. We used the following standard data-augmentation methods:

TRAIN_TRANSFORMS=transforms.Compose([
transforms.RandomResizedCrop(224),
transforms.RandomHorizontalFlip(),
transforms.ToTensor(),
normalize
])
TEST_TRANSFORMS=transforms.Compose([
transforms.Resize(256),
transforms.CenterCrop(224),
transforms.ToTensor(),
normalize
])

Note that we also tested the setting where after removing the final output layer of size 1000 is simply replaced with a single fully connected layer of size $k$ which is randomly initialized. We did not observe any change in the final accuracy.

**Training binary classifiers for codebooks:**
We want to train binary classifiers in exactly the same manner in which the multi-class classifier is trained, except here the final fully connected layer of size 1000 is replaced with a small deep neural network with 3 hidden layers of size $2000 - 1000 - 500$ and the final output layer of size 2. The weights of the small DNN are randomly initialized and we use cross-entropy binary classification loss. However, to reduce the computation time, we make use of the feature-extraction trick which we describe next.

We note that since the weights of all the previous layers except for the last three hidden layer small DNN are fixed, therefore the input to the small DNN is exactly same irrespective of the subsequent small DNN, we therefore first extract and save features along with their class labels as numpy matrices after every epoch, for a total of 150 epochs. Recall that since we use data-augmentation, therefore we need to do it for 150 epochs. We use these extracted features along with the class labels as our new training dataset. Note that for any particular dataset and model type, this computation needs to be done only once. Next, we define a 3 hidden layer binary neural-network with extracted feature size as the input size and 2 as output size. We then train this randomly initialized small binary neural-network using exactly the same hyperparameters and training schedule as described for the multi-class classifier. Using this procedure, we obtain a speed-up of around 50x-60x.

## 7.8 Dense Codes:

Another way of generating error-correcting output codes for classification known as Dense codes was proposed in [2]. Authors in [2] propose generating 10000 matrices, whose entries are randomly selected. The elements are chosen uniformly at random from $\{+1, -1\}$ and the resulting codebooks are called dense codes. Out of the 10000 random matrices generated, after discarding matrices which

do-not constitute a valid codebook, the one with the largest minimum Hamming distance among rows is selected. Note that since out of the 10000 matrices the one with the largest minimum Hamming distance is selected, therefore despite the matrices being generated randomly, the final codebook can have very high row-separation.

In our IP experiments we have provided two baselines for Dense codes. In the baseline labelled as Dense (random), we randomly generated 1000 dense codebooks and have provided the average of their min. Hamming distance. This corresponds to the expected value of the min. Hamming distance of a randomly generated dense matrix. In the baseline labelled as Dense (best out of 10k), for each experiment we generated 10000 random dense codebooks, removed the ones which do not correspond to a valid code, and from remaining chose the one with the Max. min Hamming distance. We reported the averages of 10 such runs for each $k$.

Note that [2] also proposed a way to generate random ternary codes known as sparse codes. In all our experiments sparse codes provided relatively low multi-class classification accuracy. Therefore in comparison to Dense, Hadamard and OVA, we believe that sparse-codes do-not serve as a good benchmark to compare classification accuracy.

### 7.9 Justification for $\lceil \log_2 k \rceil$

For a binary code with $k$ classes, the minimum numbers of columns ($L$) it can have, while still uniquely encoding each class is $\lceil \log_2 k \rceil$. This can be easily derived using the fact that $2^L \geq k \implies L \geq \log_2 k$. Since $L$ has to be an integer, therefore we get $L \geq \lceil \log_2 k \rceil$.

### 7.10 Scalability to Extreme Multi-Label (XML) Classification tasks:

Extreme multi-label (XML) classification refers to classification tasks where the goal is to predict a small subset of relevant labels from an extremely large set of labels. The size of the label set (i.e. $k$) can be as large as 100 million; typically the range is of the order of $10^4 - 10^7$. For such large label spaces deep neural networks (DNNs) have limited applicability as the final fully connected output layer has order $\mathcal{O}(k^2)$ number of parameters, and therefore DNN models often do not fit in the GPU memory [29]. Due to this limitation, the use of 1-vs-All type classifiers is highly predominant for XML classification tasks [48, 23, 3]. Still, 1-vs-All requires training of $k$ individual *binary* classifiers, which can be computationally quite expensive especially if $k$ is of the order of $10^6$, as a large number of binary classifiers (order of $10^6$) need to be trained. Therefore ECOC-based classifiers can very useful here. In particular, *compact* ECOC codebooks with *high error-correcting capability* can be a good substitute for 1-vs-All codebook.

In particular, ECOC-based codebooks offer three major advantages in comparison to 1-vs-All codebook: first, the training time can be significantly reduced if a compact codebook is used. Second, the prediction time will automatically reduce as the number of binary classifiers on which a data-point needs to be evaluated is much smaller. Third, as the number of required binary classifiers are small, therefore the overall model size will also be reduced significantly. Faster prediction times and smaller model sizes are important for real-world deployment.

The Greedy algorithm developed in this paper can be easily adapted to generate high-quality *compact* codebooks for XML classification tasks when $k = 10^4 - 10^6$. The main modification required is that instead of solving the vertex-coloring problem exactly, an inexact vertex-coloring solution procedure based on popular heuristics such as DSATUR [4], RLF [28] or ColPack [13] can be used. These heuristics can scale to very large graphs as their complexity scales logarithmically in the number of vertices ( equal to $k$) , i.e. $\mathcal{O}(|V| \log k)$, where $|V|$ is the number of edges [44]. Once we obtain the vertex-coloring solution, we can use the color assignment to generate columns as discussed earlier in section 7.5.1. Therefore, our greedy algorithm can scale to very large problem sizes such as XML classification tasks.

## 7.11 McCormick Inequalities

We provide a general derivation of McCormick inequalities. Consider the bilinear term $xy$ i.e. the product of two variables $x$ and $y$. Bounds on both the variable are assumed to known:
$$x^L \leq x \leq x^U$$
$$y^L \leq y \leq y^U$$
$$z = xy$$

Since, $x - x^L \geq 0$ and $y - y^L \geq 0 \implies (x - x^L)(y - y^L) \geq 0$, therefore:

$$(x - x^L)(y - y^L) \qquad \geq 0$$
$$xy - xy^L - yx^L + x^L y^L \geq 0$$
$$z - xy^L - yx^L + x^L y^L \quad \geq 0 \quad \text{as } z = xy$$
$$z \geq xy^L + yx^L - x^L y^L$$

Since, $x^U - x \geq 0$ and $y^U - y \geq 0 \implies (x^U - x)(y^U - y) \geq 0$, therefore:

$$(x^U - x)(y^U - y) \qquad \geq 0$$
$$x^U y^U - yx^U - xy^U + xy \geq 0$$
$$x^U y^U - yx^U - xy^U + z \quad \geq 0 \quad \text{as } z = xy$$
$$z \geq yx^U + xy^U - x^U y^U$$

Since, $x^U - x \geq 0$ and $y - y^L \geq 0 \implies (x^U - x)(y - y^L) \geq 0$, therefore:

$$(x^U - x)(y - y^L) \qquad \geq 0$$
$$yx^U - x^U y^L - xy + xy^L \geq 0$$
$$yx^U - x^U y^L - z + xy^L \quad \geq 0 \quad \text{as } z = xy$$
$$yx^U - x^U y^L + xy^L \geq z$$

Since, $x - x^L \geq 0$ and $y^U - y \geq 0 \implies (x - x^L)(y^U - y) \geq 0$, therefore:

$$(x - x^L)(y^U - y) \qquad \geq 0$$
$$xy^U - xy - x^L y^U + yx^L \geq 0$$
$$xy^U - z - x^L y^U + yx^L \quad \geq 0 \quad \text{as } z = xy$$
$$xy^U - x^L y^U + yx^L \geq z$$

Further, since $x \in [-1, 1]$ i.e. $x^L = -1, x^U = 1$ and $y \in [-1, 1]$ i.e. $y^L = -1, y^U = 1$, therefore:

$$z \geq -x - y - 1 \tag{15a}$$
$$z \geq \phantom{-}x + y - 1 \tag{15b}$$
$$z \leq -x + y + 1 \tag{15c}$$
$$z \leq \phantom{-}x - y + 1 \tag{15d}$$

Note that if $x \in \{-1, +1\}$ and $y \in \{-1, +1\}$, and since $z = xy$, then $z$ should also take integer values i.e. $z \in \{+1, -1\}$. However addition of constraint like $z^2 = 1$ or $z \in \{-1, +1\}$ is not required as the equations (15) are tight and ensures that $z \in \{+1, -1\}$.

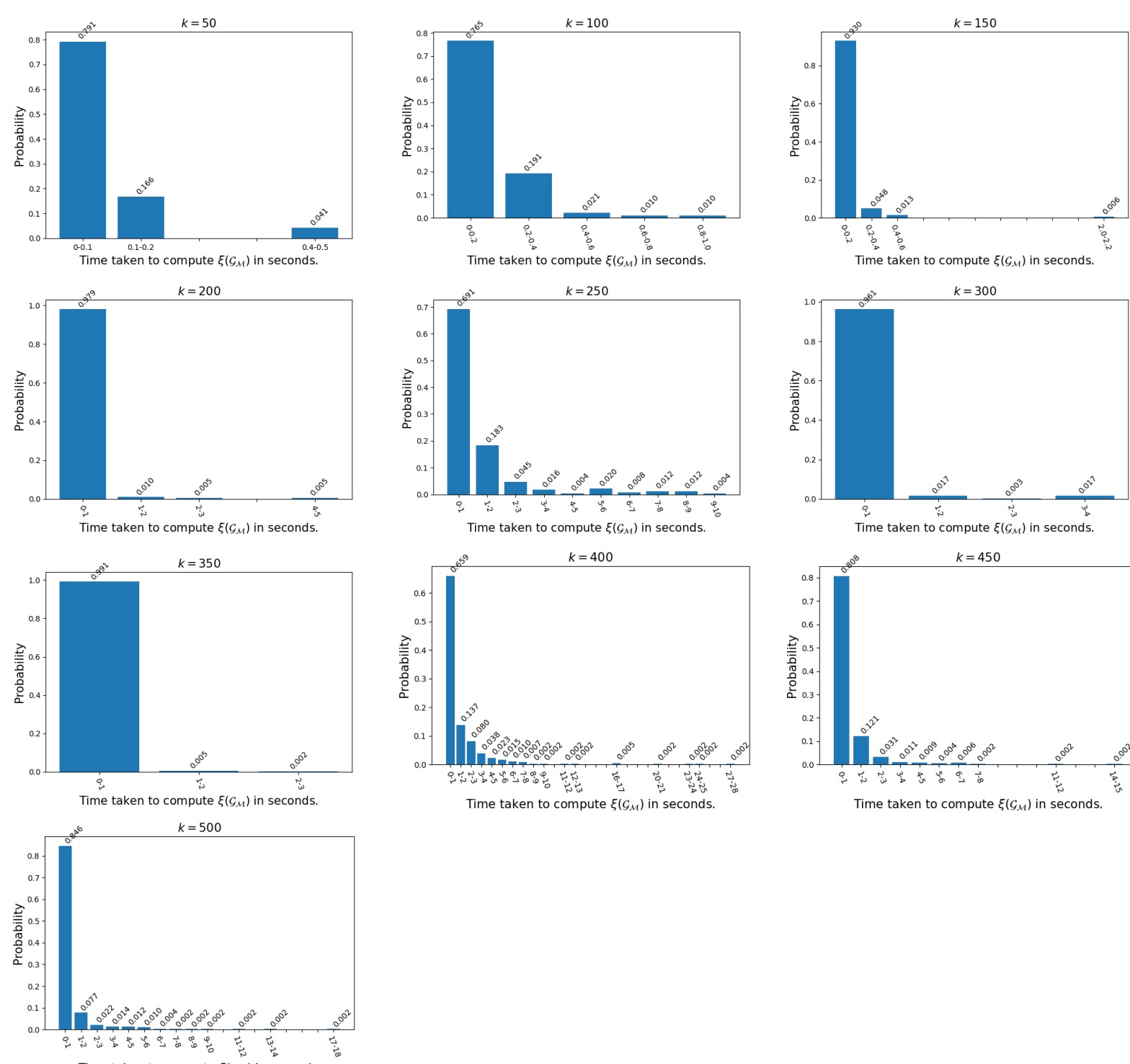

Figure 7: Vertex-coloring solve time distribution

### 7.12 Rank-1 Semidefinite Relaxation:

Any non-convex QCQP can be written in the following standard form:

$$\mathcal{Q}_1 : \min_x \quad x^T P_0 x + q_0^T x + r_0$$
$$\text{s.t.}$$
$$x^T P_i x + q_i^T x + r_i \leq 0 \qquad i = 1, \ldots, m$$

Since, $x^T P x = \text{Tr}(P(xx^T))$, we can re-write the above quadratic program as follows:

$$\mathcal{Q}_1 : \min_x \quad \mathbf{Tr}(XP_0) + q_0^T x + r_0$$
$$\text{s.t.}$$
$$\mathbf{Tr}(XP_i) + q_i^T x + r_i \leq 0 \qquad i = 1, \ldots, m$$
$$X = xx^T$$

We can now relax the above problem into convex problem by replacing the last non-convex equality constraint $X = xx^T$ with a (convex) positive Semi-definiteness constraint $X - xx^T \succeq 0$. We can now get a lower bound on the optimal value of $\mathcal{Q}_1$ by solving the following convex problem:

$$\min_x \quad \mathbf{Tr}(XP_0) + q_0^T x + r_0$$
$$\text{s.t.}$$
$$\mathbf{Tr}(XP_i) + q_i^T x + r_i \leq 0 \qquad i = 1, \ldots, m$$
$$X \succeq xx^T$$

The last constraint $X \succeq xx^T$ is convex and can be formulated as a Schur complement.

$$\min_x \quad \mathbf{Tr}(XP_0) + q_0^T x + r_0$$
$$\text{s.t.}$$
$$\mathbf{Tr}(XP_i) + q_i^T x + r_i \leq 0 \qquad i = 1, \ldots, m$$
$$\begin{bmatrix} X & x \\ x^T & 1 \end{bmatrix} \succeq 0$$

The above optimization problem is an SDP and is known as the Rank-1 SDP relaxation of the non-convex QCQP. The optimal value of this SDP is a lower-bound on the optimal value of the non-convex QCQP, i.e. $\mathcal{Q}_1$.