# OpenReview forum: "Scalable design of Error-Correcting Output Codes using Discrete Optimization with Graph Coloring"
_NeurIPS.cc/2022/Conference — NeurIPS 2022 Accept_

### Official Review · Reviewer_t9cy · 2022-07-11

**Rating:** 6
**Confidence:** 2
**Soundness:** 3 good
**Presentation:** 3 good
**Contribution:** 3 good

**Summary:**

To solve the problems of prior works (e.g., limited to codebooks with small number of rows (classes) or columns), the authors develop a codebook design approach based on a Mixed-Integer Quadratically Constrained Program (MIQCP). Correspondingly, a novel and efficient algorithm to solve large instances of the MIQCP is proposed, that is, it incrementally increases the codebook size by adding columns to maximize the gain in error-correcting capability.

**Questions:**

The detailed questions follow:

1. This paper emphasizes that the previous approaches have the limitation ``the size of the codebook (i.e. number of columns L) is small".
How do the authors judge whether $L$ is small? If $L<2k$, is $L$ small? To distinguish your work from previous works, can you give a sensible judgment criterion?

2. Line 142: this paper emphasizes ``the error correcting capability ... by increasing the number of columns". Does that mean that bigger $L$ is better? Does a very large $L$ have a negative effect on the classification performance? Is there a limit on the upper bound of $L$ in this paper?

3. Given the codebook $\mathcal{M}$, the training data and a test example $x$, how are the binary classifiers $f_1(x),\ldots,f_L(x)$ determined or obtained?

4. Are the parameters $\rho_1$, $\rho_2$ and $\gamma$ in $\mathcal{IP}$1 and $\mathcal{IP}$2 given in advance? Does the choice of these parameters affect the classification performance? Do they affect the optimality of the algorithm? Moreover, in section 5, the specific values of the parameters $\rho_1$, $\rho_2$, and $\gamma$ is not clear.

5. For $\tilde{l}\in\{1,2,3,4\}$, the authors give the upper bounds of $\kappa_{\tilde{M}}$ in Theorem 3. For any $\tilde{l}$, can you give a general expression with the detailed proof?

6. For the computation in Table 1, is $\mathcal{M}$ given in advance? What is the exact structure of $\mathcal{M}$? Furthermore, is it possible that $\mathcal{G}_{\mathcal{M}}$ is not sparse?  If so, will the computation be slow?

7. How to ensure that the proposed algorithm is near-optimal? Please give a detailed explanation. Moreover, is the codebook obtained by the proposed algorithm very close to the solution of $\mathcal{IP}$1?

8. In this paper, the $L$ is given in advance. Can an optimal $L$ in some sense (e.g., minimizing the classification error) be obtained? It would be great if the authors could develop an algorithm or give a theoretical guidance to derive an optimal $L$.


**Limitations:**

Limitations are discussed above. I do not see any societal impact.

**Strengths And Weaknesses:**

This paper is very well-written and easy to read. For the large multi-class problem instances, this paper propose a novel approach to effectively solve the codebook design problem.

---

> ### Author Response · Authors · 2022-08-01
> **Response to reviewer t9cy questions 5-8.**
>
> We thank the reviewer for their comments and feedback. Please see our answers to questions 5-8 below:
>
> 5) For any given $\tilde{l}$, the bounds on $\mathcal{K}_{\tilde{\mathcal{M}}}$ can be easily obtained using the procedure described on page 3 in the supplementary material.  A generic analytical expression is difficult to characterize as the  number of codewords of size $\tilde{l}$ grows with $2^{\tilde{l}}$, however we can provide it as an algorithm. Further, it is also preferred to keep $\tilde{l}$ small so that the number of bi-linear terms in $\mathcal{IP}2$ remains small to maintain tractability.
>
> 6) In Table 1, we have provided the average computation times of solving the graph coloring problem for the graph $\mathcal{G}_{\mathcal{M}}$ corresponding to the codebook $\mathcal{M}$ obtained at each iteration of Algorithm 1. The exact structure of $\mathcal{M}$ is hard to define, however for most of the cases $\mathcal{G}_\mathcal{M}$ is sparse and the probability of encountering scenario where the computation time is large is very low. On page 11 in the supplementary material, we have provided the entire distribution for graph-coloring computation times for different $k$.
> Further we would like to point out that with small modification our approach is scalable to even extreme classification settings when $k$ is of the order of $10^4-10^6$. We refer the reviewer to section 7.8 on page 9 in the supplementary material.
>
> 7)  The proposed algorithm aims to solve or generate a near-optimal solution to the codebook design problem i.e. $\mathcal{IP}1$.  Given the discrete, constrained and non-convex nature of $\mathcal{IP}1$ we cannot use conventional convex-optimization techniques to determine optimality. Further we do not know what the exact optimal objective function value $f^*$ for any given $\mathcal{IP1}$. If suppose the exact optimal objective function value $f^*$ would have been known, then we could have easily computed the difference between the solution of algorithm 1 (denoted as $f_{best}$) and $f^*$, i.e. $\frac{f^* -f_{best}}{f_{best}}$ , also commonly referred to as the optimality gap. This gap provides a principled way to evaluate the solution quality. Now in the absence of $f^*$, what one can try to do is compare $f_{best}$ against a valid upper bound (denoted $UB$) to any feasible solution of $\mathcal{IP}1$. As UB is a valid upper bound therefore we can write the following:
> $f* \le UB$
> Since $f_{best}$ is feasible solution, therefore:
> $f^*-f_{best} \le UB - f_{best} \implies \frac{f^*-f_{best}}{f_{best}} \le \frac{UB - f_{best}}{f_{best}} $.
> Essentially using $UB$ we are able to upper bound the optimality gap. Therefore if $\frac{UB - f_{best}}{f_{best}} $ is small then it is also guaranteed that $\frac{f^*-f_{best}}{ f_{best} }$ is small. Note that one would prefer $UB$ to be close to $f^*$ or relatively tight.
> The main question which arises is from where to get $UB$.
> For  $\mathcal{IP}1$ we can thankfully use Plotkin’s Bound (denoted $PB$) as it analytically upper bounds the error-correcting property of any binary code of size $k \times L$ as follows:  $PB = \frac{kL}{2(k-1)}$.
> In table 2, we provide these gaps for different values of k and also compare with the approach of Gupta and Amin 2021. As one can note that for small problem instances when $k \le 50$, we either match or perform slightly better than the Gupta and Amin approach.  For large problem sizes i.e. $k \ge 100$, we maintain small gaps (or near-optimality), while the Gupta and Amin approach cannot even generate a feasible solution for these cases.
> We would also like to highlight that most papers in literature on ecoc have not provided any principled analysis of their optimization solution quality.
>
> 8) As we mentioned earlier in response to question 2, one of the interesting and unique characteristics of our approach is that  in each iteration  it adds $\tilde{l}$ columns to an existing codebook. Therefore, one can generate a small codebook, train the codebook, evaluate the classification performance and generate more columns (via solving $\mathcal{IP}2$), train the new columns, add them to existing columns and then re-evaluate the classification performance and keep doing this till there is almost no improvement in the classification accuracy. This therefore solves the issue of choosing $L$ or pre-defining an optimal $L$, which will vary with the dataset. To the best of our knowledge, we are the first to propose such a flexible approach which alleviates the issue of predefining $L$.

---

> ### Author Response · Authors · 2022-08-01
> **Response to reviewer t9cy questions 1-4.**
>
> We thank the reviewer for their comments and feedback. Please see our answers to questions 1-4 below:
>
> 1) We would define that $L$ to be  small (and a major limitation) when L is roughly 4 times smaller than k,  provided k is also of the order of 10-500.  For example consider the paper of: Martin et. al.,Error-correcting factorization. IEEE Transactions on PAMI 2018. In this paper, the authors propose a method for designing a codebook using continuous relaxation of the discrete codebook design optimization problem  with quadratic constraints and a more refined objective function. They provide experiments for $k$ in the range of 10-50. However, the number of columns $L$ in their codebooks are very small as shown in figure 4 of their paper. For $k = 36$, they use $L = 6$ and for $k = 50$ they use $L = 8$. Clearly, their $L$ is small and as a consequence of small $L$, their approach is not able to consistently match (or outperform) commonly used standard codebooks, let alone multi-class CNNs. They only provide experiments for  $k \le 50$. Further, they do not provide optimality gaps or any discussion on whether their solution approach converges when  $L$ is increased for these small values of $k$.
> On the other hand our approach easily scales to $k = 500$, as our approach adds columns in each iteration, therefore our approach does not face any issues when $L$ is large even if $L$ is say twice the size of $k$. Interestingly, our approach does not even need to pre-define $L$, as discussed subsequently.
>
> 2) Yes, a large $L$ is preferred because then a codebook can have a better error-correcting property. A simpler way to look at this is through the Plotkin’s bound (provided in the paper in line 91), which upper bounds the error-correcting property and is equal to $Lk/(2(k-1))$.
> There is no negative effect on having a large $L$ on the classification performance, however for extreme classification cases when k is of the order of $10^4-10^6$, a smaller $L$ would be preferred for computational reasons.
> $\textbf{Flexibility of our approach in choosing } L $
> Defining an upper bound over $L$ or choosing $L$ for past ecoc design approaches is hard and problematic as right $L$ may highly vary from dataset to dataset. For all past ecoc design approaches, to the best of our knowledge, one needs to pre-define $L$ in order to solve the optimization problem, and if the observed performance is bad then one may increase $L$, resolve the codebook design problem, and then need to re-train all the columns in the new codebook. Clearly this entails a very large computation burden as the computation effort spent on previous codebooks is completely wasted.
> In a sharp contrast, due to the additive column nature of our approach choosing $L$ is not a problem thus entirely alleviating the issue of pre-determining $L$. With our approach, one can start off with a small $L$, train the columns, evaluate the classification performance and then  obtain another $\tilde{l}$ columns by solving $\mathcal{IP}2$, training them and add them to the existing columns, re-evaluate the classification performance and continue to do so till there is no improvement in the classification performance. Therefore, importantly the computation effort spent in training previous columns is not wasted and one can stop anytime if the improvement in the classification performance is negligible. Apart from scalability, this is another major practical advantage of our approach.
>
> 3) The binary classifiers $f_1(x) \dots f_l(x) $ are trained by converting the multi-class data to two classes according to the entries +1 and -1 in the corresponding columns of the codebook $\mathcal{M}$. Model architecture level details are provided in section 7.5 in the supplementary material.
>
> 4) Yes, the parameters $\rho_1,\rho_2 \text{ and } \gamma$ in $\mathcal{IP1}$ and $\mathcal{IP}2$ are given in advance and are defined with respect to $k$. The choice of these parameters can affect the classification performance, however in our experience and from most of the literature the final accuracy is not highly sensitive to $\rho_1 \text{ and } \rho_2$, as long as they are defined in a reasonable range. The value of $\gamma$ is to be chosen such that a very hard balanced column criteria is to be avoided as it negatively impacts the classification accuracy. Its choice also depends on the type of the binary classifier which one intends to use for each column.
> Importantly, the optimality of our algorithm is not affected because in each iteration we are able to solve $\mathcal{IP}2$ optimally, thanks to the graph-coloring bound (Theorem 3). In the final version we will provide discussion on how to choose these parameters including their appropriate ranges and specific values for different $k$.

---

### Official Review · Reviewer_FwJk · 2022-07-11

**Rating:** 6
**Confidence:** 4
**Soundness:** 3 good
**Presentation:** 3 good
**Contribution:** 2 fair

**Summary:**

The paper proposes a way to design dataset-independent error-correcting codebooks with good error-correcting properties.

The suggested approach (naturally) formulates a *discrete* optimization problem. Rather than solving the problem using continuous relaxations (like most of the existing literature), the authors follow a recent work *[Gupta & Amin 2021]* and use Integer Programming (IP) to solve the optimization problem. They solve that problem using a *greedy* algorithm that solves smaller IP problems until a codebook with the required amount of columns (bits) is obtained.
The algorithm proposed here scales better with the number of columns and yields good codebooks even when other algorithm become infeasible.

The experimental setup establishes that the resulted codebooks are nearly optimal (in the sense of their minimal Hamming row distance) and are useful for classification setups.

**Questions:**

1. **Fixable readability issues.** Personally, I found Pages 4-6 difficult to read and hard to follow. In Page 6 I already felt lost and had to stop, re-read, and think for a few minutes until I realized what is happening.
At this point it's probably trivial for the authors, but as a reader I think the connection to the graph coloring problem should be presented more carefully.
A few specific issues:
    - I had a hard time understanding why would different vertices get the same codeword (Lines 209-210). It is worth stressing out that the discussed codewords are only the ones of length $\tilde{l}$ that are being concatenated to the existing (full) codewords.
    - Proposition 2 is presented as a direct consequence of the two observations above it, but it is not completely clear how are they related. Also unclear why is this Proposition important or interesting.
    - The upper bound in Theorem 3 is hard to understand.
    - Unclear how $\tilde{l}$ is chosen/tuned in Step 5 of Algorithm 1.

1. In Lines 300-301 and Table 4 it is reported that multiclass (Softmax) CNNs have a comparable performance to that of the Greedy codebook. If I understand correctly, this happens even though the traditional multiclass approach only requires one network while the ECOC approach here is implemented with $L$ networks.
If that is indeed the case - why should we use the Greedy codebook?

1. In Tables 5-7, it is reported that the Greedy codebook outperforms the multiclass approach significantly. Again, it seems like the greedy ECOC approach uses $L$ networks while the multiclass approach uses only a single network. If that is indeed the case - is this comparison even fair?

1. In Table 1:
    - Are these averages of several runs? How many? What is the STD?
    - How should we expect the time to increase with $k$? What will happen if we use this approach for $k>10K$ like in extreme classification papers? *[See for instance "Learning compact class codes for fast inference in large multi class classification" by Cisse et al. 2012]*

1. It seems odd that in Table 4 the performance of the greedy algorithm is the exact same as the performance of [Gupta and Amin 2021]. Is there a reason for it?

1. In the classification performance experiments (Section 5.1): Which ECOC decoding scheme is used? Specifically, which decoding scheme is used for 1-vs-1? Do you ignore the prediction magnitudes?

**Limitations:**

Addressed properly.

**Strengths And Weaknesses:**

# (A) Strengths

1. **Proposed algorithm improves upon an existing method.** The paper proposes an approach that solves the scalability issues of a recent work by *[Gupta and Amin 2021]*, by using graph coloring techniques instead of edge cover ones.
 Both papers are rather unique in the sense that they tackle the codebook design problem using discrete methods instead of continuous relaxations, providing another perspective on designing codebooks.


----


# (B) Weaknesses

1. **Missing important/natural baseline for classification performance in Tables 4 and 7.** I would expect the experiments to compare to the performance of (truncated) Hadamard codebooks *[See for instance "Fix your classifier" by Hoffer et al. 2018]*. Hadamard codebooks should have almost optimal error-correcting properties and may be decent competitors to the suggested approach.
Since the proposed approach here is dataset-independent, I cannot immediately see why this approach would be favorable over Hadamard codebooks.


1. **Missing baseline for error-correcting capabilities in Table 2.** I believe that Dense random codebooks should also be compared to the ones created using the proposed Greedy algorithm, since random codebooks usually exhibit good error-correcting capabilities (and specifically, a good minimum distance when $L\ge k$).


1. **Some missing/unjustified "goals" in main Problem Formulation (Section 3).**
     - The authors claim that *"a good column separation is desirable in order to avoid correlations between the resulting hypotheses"*. I agree.
However, one should also consider the separation between the set of columns and the set of their binary complements. For instance, if there exists a pair of columns in the matrix such that one of them is the exact complement of the other, then they induce the same binary classification problem and will yield two completely correlated classifiers.
Ideally this should be part of the formulation and the subsequent integer programs. But at least it should be mentioned and/or its absence should justified.
     - The authors also claim that *"every column in the codebook should result in a balanced binary classification problem"*.
While I agree that this may help a codebook's error-correcting properties, its effect on an ECOC multiclass classification scheme is not *that* simple.
On the one side, imbalanced binary problems might suffer statistical drawbacks. But on the other side, imbalanced problems (in the sense of few classes vs. the rest) can be *much* easier to solve, especially when the basis classifiers are not very complex (e.g. linear models). This phenomenon can explain the historic success of OVA when compared to other ECOC schemes *[as was already pointed out by Allwein et al. 2000; see also Rifkin & Klautau 2004]*. Another paper *[Evron et al. 2018]* also pointed out that the column imbalance can actually be beneficial for learning.


---

# (C) Minor remarks

I don't expect a response on these issues but I advise fixing them.

1. In Line 307 the authors say that the computational expense of training $L$ separate neural networks is an inherent limitation of ECOC classifiers, but there were already several of other papers that used ECOC as an output layer of a **single** neural network *[e.g. "Beyond One-hot Encoding" by Rodriguez et al. 2018]*.

1. References to the supplementary material ("SI") should at least refer to specific sections.

1. I believe the models used for the experiments should be briefly explained in the main manuscript and not only in the supplementary material.

1. Typos:
   - In many places (e.g., line 200&201) the authors write "atleast" or "atmost". I believe the correct forms are "at least" and "at most".
   - Line 188: "pairs decides" => "pairs decide"
   - Line 192: redundant parenthesis ")"
   - Line 217: "than size" => "than the size"
   - Line 240: unclear what the double dashes mean.

1. Generally - it's unclear why you almost always consider codebooks where $k<L$.

1. Should probably have a more thorough discussion on the relations to *[Gupta and Amin 2021]* and the differences between the two papers.

---

> ### Author Response · Authors · 2022-08-01
> **Response to reviewer FwJk questions 1-6**
>
> We thank the reviewer for their questions. Please see our answers to questions 1-6 below:
> 1) Readability issues:
>       - We will improve the discussion on pages 4-6 so that it is easier to follow including the connection to graph coloring. We will stress that the codewords being added are of length $\tilde{l}$.
>       - Proposition 2  provides a gentle introduction to graph coloring using the two observations. The first observation highlights that to increase the hamming distance of the codebook by 1, one does not need to focus on all rows pairs but instead only on row-pairs in the set $\mathcal{E}_{\mathcal{M}}$. The second observation highlights that codewords are distinct.
>       - We will add discussion to more clearly explain the upper bound in Theorem 3.
>       - Regarding choosing the value of $\tilde{l}$, we refer the reviewer to lines 244-248 on page 7 in the paper.
>
>      We will add discussion to further clarify all the above points.
>
> 2) In Table-4 the ecoc approach uses $L$ networks and provides slightly better accuracy over multi-class CNN. The main goal of this table is to simply verify the ecoc theory on small class datasets. In the introduction we have provided references which highlights the numerous applications of ecoc. Importantly, the Gupta and Amin 2021 paper has shown that ecoc based classifiers provide much better robustness to adversarial perturbations in comparison to multi-class CNNs. The codebook design approach of most papers in literature including Gupta and Amin 2021, do not provide a scalable solution algorithm, particularly when L is large, therefore, the main focus of this paper is to provide a scalable solution to the codebook design problem along with a principled evaluation of the solution quality in terms of the optimality gap.
>
> 3) On large class datasets, Tables 5-7 do provide a fair comparison as here the muti-class CNN and the ecoc binary classifiers are all trained using transfer learning from an ImageNet-1k model where only the weights of the last layer are allowed to change. Please refer to lines 315-323 on page 9 and section 7.5.2 on page 8 of the supplementary material.
>
> 4) Yes, in table 1 we have reported the average time taken to solve each graph-coloring problem. These are averages of all the graph coloring instances that will be solved for a particular $k$ to generate a codebook of size $L=2k$ with $\tilde{l} =2$. For example,  for k=50 this corresponds to the average of ~ 50 runs and for K=500  this corresponds to the average of ~ 500 runs of the graph coloring instances. We have in fact provided the entire histogram for each k on page 11 in the supplementary material. Note that  with very low probability $(<1\%)$ one can encounter  scenarios for which the computation time is considerably higher than the mean thus skewing the standard deviation. Therefore to avoid any confusion we have reported the entire distribution. For these long-tailed distributions, we believe that std would mis-represent the actual computation times given that the mean is close to zero and the de-facto assumption of normal distribution doesn’t hold here.
>
>    We have discussed the extreme classification case in particular the eXtreme Multi-Label (XML) classification on Page 9 in section 7.8 in the supplementary material. We are in fact currently extending this work to XML settings. We will include the related Cisse et al. paper in the introduction.
>
> 5) The reason in table 4 the Gupta and Amin approach has the same accuracy as our greedy approach is because both the approaches generate optimal codebooks with the same objective function value. In fact for CIFAR10, Gupta and Amin in their paper reported an  accuracy of 76.25% which is clearly quite worse than our 95%. To provide a fair treatment to the approach of Gupta and Amin, we used our training procedure to re-train their codebooks instead of values reported in their paper. We have pointed this out in the footnote on page 8.
>
> 6) No, we did not ignore the prediction magnitudes at the time of decoding.  We use a class score based decoding scheme. We  compute a  score for each class by carefully adding the output score of each  binary classifier logit depending upon whether the entry of that class (or row) has +1 or -1. Normalization is not required for binary codes as each row of a binary code has the same number of entries. However we were careful with ternary codes that appropriate normalization is done in particular for sparse codes. 1-vs-1 (also a ternary code) is balanced in the sense that each row has the same number of zero entries hence normalization is not required. In fact we observed that score based decoding consistently provides slightly better accuracy over Hamming decoding. Further Hamming decoding has issues when working with ternary codes as pointed out in the paper by Pujol et al. Discriminant ECOC: A Heuristic Method for Application Dependent Design of Error Correcting Output Codes.

---

> ### Author Response · Authors · 2022-08-02
> **Response to reviewer FwJk on Weaknesses 2 and 3**
>
> 2) Missing baseline for error-correcting capabilities:
> We have updated the paper to include comparison with Dense codebooks in Table 2.
> We have added two baselines,
> **Baseline 1** - Dense (random):
> We randomly generated 1000 dense codebooks and have provided the average of their min. Hamming distance. This corresponds to the expected value of the min. Hamming distance of a randomly generated dense matrix.
> **Baseline 2** - Dense (best out of 10k): (Allwein et al. paper)
> In this baseline, for each experiment we generated 10000 random dense codebooks, removed the ones which do not correspond to a valid code, and from remaining chose the one with the Max. min Hamming distance. We report the averages of 10 such runs for each $k$.
>    We note that for all $k$ our  Greedy codebooks easily outperform Dense codebooks. For $k < 50$,  baseline 1 the gap is well beyond 50% and can be as high as 88% for smaller k and even for  baseline 2 the gap is well beyond 30% and can be as high as 44\%. The gap of Greedy codebooks is almost 3 times smaller than the dense codebooks.
> For $ 50 \le k \le 300$, the gap of dense codebooks (even with baseline 2) is twice that of Greedy codebooks.  For $k \ge 350 $, dense codebooks do show low gaps, but are still worse than Greedy  codebooks.
> 3) Missing/unjustified goals in problem formulation (section 3):
>   **Avoiding complementary columns:**
>   We agree with the reviewer that complementary columns should be avoided as they essentially correspond to the same binary classifier.
> We would refer the reviewer to line 93 where we have stated that complementary columns should be avoided. To ensure this the hamming distance between any two columns is upper bounded with $\rho_2$, where $\rho_2$ should be less than $k-1$.
> In the integer programming (IP) formulation ($\mathcal{IP}1$), constraint 1c ensures that the Hamming distance between any two columns is upper bounded to avoid complementary columns.
>   **Balanced column criteria:**
>    - Firstly, our main motivation to include the balanced column criteria is to show that our IP formulation is generic and flexible, that this criteria can be easily incorporated and the proposed greedy algorithm can handle this criteria if desired. In particular, in our IP formulation this criteria can be easily controlled using the parameter $\gamma$. If one wants to enforce this constraint as a hard (tight) or soft constraint one can choose the value of gamma accordingly. For instance, for a dataset with the same number of training data-points in each class, and if $k$ is even then to enforce a very hard balanced column criteria $\gamma$ can be set to $0$, so that each binary class receives exactly the same number of $k/2$ classes (i.e. their datapoints). On the other hand if enforcing this criteria is not desirable at all then  gamma can be set to a  trivial value of $k-1$ to simply ensure that a column does not have all +1s or all -1s. Further one is also free to choose some intermediary value as well.
> Further we did experiment with a very hard balanced column criteria setting where we set gamma to very low values such that only near 50%-50% splits are allowed, and in this setting we actually observed a slight reduction in the final accuracy. Therefore, admittedly, we are also of the opinion that a very hard balanced column criteria should be avoided, however, if desired for some  reason, then our formulation and solution algorithm is flexible to incorporate it.
> To avoid this confusion, we will paraphrase the balanced column criteria, so that it does not portray that we promote a hard balanced column criteria.
>    - Secondly, we completely agree with the reviewer's characterization that training the resulting column (balanced or imbalanced) heavily depends on the complexity of the binary classifier which one decides to use. Imbalance columns are easier to learn for simple (linear) models however can be challenging for complex models like DNNs (requiring to choose the training loss function carefully). Further, in real-world settings, some classes out of $k$ classes, may have much less training data, in those scenarios column imbalance is further exacerbated. For a highly imbalanced column with DNNs, overfitting can be an issue thus adversely affecting final classification performance.
>    For the above reasons, we decided to use an intermediary value of gamma and avoided hard enforcement of balanced columns.  However, as we already mentioned above, our IP formulation and greedy solution algorithm are flexible to (or not to) enforce this criteria depending upon the choice of binary classifier which one plans to use down the line. Even though we have cited all the three papers i.e. Allwein et al. 2000,Rifkin & Klautau 2004 and Evron et al. 2018 in the introduction, we will refer to them again in the balanced column criteria discussion.  We will add more discussion regarding the subtle details of the balanced column criteria in the paper.

---

> ### Author Response · Authors · 2022-08-02
> **Response to reviewer FwJk on Weaknesses 1**
>
> We thank the reviewer for their comments and feedback. Please see our response below:
> **Hadamard codebooks** (Baseline):
> In the updated version of the paper, we have added the comparison with (truncated) Hadamard codebooks in Tables 3-7. Hadamard codes  do provide competitive classification performance but only when the full Hadamard code of length $2^T-1$ (where T = $\lceil \log_2(k) \rceil$) is used. Note that the  first column is removed as it contains all 1s. The fixed length of Hadamard codes is a major issue and  severely limits its applicability to large class datasets.
>
> First, consider the small class datasets like MNIST and CIFAR 10 (with $k =10$). Hadamard code here will be of size $10 \times 15$. With only first 5 columns, i.e. $L=5$, Hadamard code does not provide satisfactory performance, whereas Greedy codebooks are able to provide high performance even with  $L=5$.
> Although, when $L$ is increased to $L=10$ or even $L=15$, then Hadamard codes do provide competitive accuracy. But due to the fixed number of columns, one cannot go beyond $L=15$ with a Hadamard code. While with Greedy codebooks, one can try to improve accuracy by adding columns beyond $L=15$.
>
> In the case of medium sized datasets (in terms of $k$), say Caltech101, with $k=101$, the performance of Hadamard code with $L=50$  is much worse in comparison to Greedy codebooks. However with  its full length $L=127$,  Hadamard code provides competitive accuracy.
>
> Finally for large class dataset like Caltech256, (with $k=257$), Hadamard code is easily outperformed by Greedy codebooks when $L= 100$ or even $L=200$. Although, when $L$ becomes really large, i.e. $L \ge k$, i.e. for $L=300$ Hadamard code provides high accuracy. We also evaluated the accuracy of the entire Hadamard code with $L=511$, although there is not much gain in accuracy from $L=300$ to $L=511$.
>
> The above results indicate that Hadamard codes can provide high accuracy but at the cost of training a large number of columns. Therefore the use of Hadamard codes entails a large computational burden. On the contrary, our Greedy approach generates compact codes which can provide accuracy similar to Hadamard codes but with a considerably small number of columns. This becomes particularly important when working on datasets in the extreme classification regime where $k $ is of the order of $10^4-10^6$. In extreme classification, use of compact codes preferably of length much smaller than $k$ is extremely desirable. In such settings Hadamard codes are clearly not tractable. On the other hand our approach is easily scalable and can be applied to such settings. We have discussed the extreme classification case in section 7.8 on page 9 in the supplementary material.
>
> An interesting point which the reviewer raises is that of dataset-dependence or data-distribution. In some aspects, our approach is dataset-depend as one does not need to pred-define a fixed $L$.
> For almost all past ecoc design approaches, to the best of our knowledge, one needs to pre-define L in order to solve the optimization problem, and if the observed performance is bad then one may increase L, resolve the codebook design problem, and then need to re-train all the columns in the new codebook. Clearly this entails a very large computation burden as the computation effort spent on previous codebooks is completely wasted.
> In a sharp contrast, due to the additive column nature of our approach, predefining $L$ is not required. With our approach, one can start off with a small $L$, train the columns, evaluate the classification performance and then  can add more columns after training them  to the existing columns and continue to do so till there is no improvement in the classification performance. Most importantly the computation effort spent in training previous columns is not wasted and one can stop anytime if the improvement in the classification performance is negligible.
> In the paper by Martin et. al. Error-correcting factorization, PAMI 2018, authors also need to solve their data-distribution informed codebook-design problem with different L to achieve good performance. Our formulation and solution approach can be adapted to their  data-distribution objective.
>
> In comparison to the “Fix your classifier” paper by Hoffer et al. 2018, (which we will cite), our approach is quite different. Hoffer et al. propose to use Hadamard code in the final layer. On the contrary, we in our transfer-learning experiments (in Tables 5-7), propose generating the multi-class classifier using separately trained binary classifiers through an error-correcting code. Most importantly, we achieve a 1-2% consistent gain in accuracy over a multi-class CNN which has also been trained using transfer-learning, when only the weights of the last-layer are allowed to change. This is another contribution of our paper, i.e. in the transfer learning setting, the accuracy of the classifier can be improved further by 1-2%.

---

> ### Comment · Reviewer_FwJk · 2022-08-07
> **Response for the authors**
>
> I thank the authors for their detailed response and answers.
>
> Several remarks regarding the responses and the revisions:
>
> 1. The authors wrote *"For almost all past ecoc design approaches, to the best of our knowledge, one needs to pre-define L in order to solve the optimization problem, and if the observed performance is bad then one may increase L, resolve the codebook design problem, and then need to re-train all the columns in the new codebook..."*.
>
>     I feel like I should refer the authors to [*Discriminant ECOC: A Heuristic Method for Application Dependent Design of Error Correcting Output Codes*; Pujol et al. 2006] and [*An incremental node embedding technique for error correcting output codes*; Pujol et al. 2008]. Obviously, the paper reviewed here is more scalable than these other two, but this paper is certainly not the first to propose a method that deal without a predefined $L$. These papers should probably be cited in that context.
>
> 1. I agree with almost everything the authors wrote in their response regarding the baselines I suggested. I still think these baselines improved the revised paper. For instance, most experiments used $0.5k \le L \le 2k$, and so, comparing to Hadamard codes (where $L=2^{(\lceil \log_2 k \rceil)}-1 < 2k$) felt fair. In extreme classification settings this of course becomes infeasible.
>
> 1. In the robust row of Table 5, I believe the Dense method should also be bold.
>
> 1. Notice that the tables no longer fit the template's margin. This should be done before publication.

---

> > ### Author Response · Authors · 2022-08-09
> > **Closing response to reviewer FwJk**
> >
> > We thank the reviewer for their subsequent follow-up comments. In the final manuscript we will carefully include all the comments and revisions which the reviewer has suggested in this thread. We will cite Pujol et al 2006 and Pujol et al. 2008 papers in the appropriate context, highlight the Dense method in the robust row in Table 5. As the final version allows an additional page, therefore we will ensure that the revised tables with the additional baselines fit the template's margin. We will also include other changes suggested in the original review.
> > Thank You.

---

> ### Comment · Reviewer_FwJk · 2022-08-07
> **(my) Final decision after rebuttal**
>
> Following the rebuttal process, I decided to increase my score from 4 to 6.
>
> I believe that the paper is solid and rather interesting. It improves a work by [Gupta and Amin 2021] that tackled the codebook design problem using an integer programming approach.
>
> Personally, I find the proposed method to be somewhat complicated and hard to work with or build upon.
> The empirical evaluation demonstrates small (but statistically significant) improvements over *much* simpler codebooks (e.g., Hadamard or Random dense).
>
> However, the paper *does* make a nice progress in the otherwise "saturated" area of designing ECOC codebooks for classification, which is studied for almost three decades now.

---

### Official Review · Reviewer_WaNX · 2022-07-15

**Rating:** 8
**Confidence:** 4
**Soundness:** 4 excellent
**Presentation:** 4 excellent
**Contribution:** 4 excellent

**Summary:**

This paper is about generating high-quality error-correcting output codes for multi-class classification. Based on the concept of Hamming distance of a codebook, it formulates this problem as solving a non-convex mixed integer quadratic constrained program. In addition to converting the bilinear constraints to linear ones, it proposes an iterative greedy approach and cleverly views the resulting sub-problem as a graph-coloring task, leading to an efficient solution. Experimental study on several datasets shows the computational advantage and effectiveness of the proposed work.

**Questions:**

This is an overall excellent work. Meanwhile, the following issues can be addressed.

1. This work mentioned in several places that the proposed greedy approach is a "near-optimal" solution to the codebook design problem. It will be better to give a clear definition or criterion on the "near-optimality."

2. How to set the value of $\rho_1$ and $\rho_2$ in Eq.(1c)? What is their impact to the quality of the obtained codebook? This could be discussed in the experimental study.

3. In Line 127, it is mentioned that "Since $L \approx k.$" Where is this condition from? Please clarify.

4. It will be good to provide a formal computational complexity analysis for Algorithm 1.

5. In the last line of Theorem 3, it shows that $\kappa_\tilde{\mathcal M}\leq\kappa_{\mathcal M}$ if $\xi({\mathcal G}_{\mathcal M})\geq{5}$.

Combining this result with Lemma 2, does this mean actually $\kappa_\tilde{\mathcal M}$=$\kappa_{\mathcal M}$ in this case? Please clarify.

6. The improvement over other methods in comparison seems to be marginal. Can the improvements pass the statistical significant test?

**Limitations:**

The limitations and potential negative societal impact are discussed in Section 6.

**Strengths And Weaknesses:**

Strengths:

1. Error-correcting output coding (ECOC) is an important mechanism in many tasks including multi-class classification. An advance in achieving high-quality large-scale ECOC design has both theoretical and practical significance.

2. The proposed approach is neat, clever and elegant, leading to a simple and efficient solution.

3. Experiments demonstrate the efficacy of the proposed algorithm.

4. The paper is well organised and presented.

Weaknesses:

1.  Some claims can be better clarified or justified.

2. The improvement seems to be marginal with respect to other multi-class classification methods.

---

> ### Author Response · Authors · 2022-08-01
> **Response to reviewer WaNX**
>
> We thank the reviewer for their comments and feedback. Please see our answers to questions 1-6 below:
>
> 1) We will clearly define that by a "near-optimal" solution to the codebook design problem i.e. $\mathcal{IP}1$, we mean how close is the objective function value of the solution obtained with algorithm 1 (denoted $f_{\text{best}}$), to the optimal objective function value $f^*$.  And since $f^*$ is unknown, we measure the optimality gap  using Plotkin’s bound which is a valid upper-bound to $\mathcal{IP}1$.
>
> 2) The values of $\rho_1$ and $\rho_2$ lower bounds and upper bound the Hamming distances between any two columns, where $\rho_1 \ge 1$ to avoid same columns and $\rho_2 \le k-1$ to avoid complementary columns. This is mainly to ensure that columns are not correlated. In our experience and from most of the literature the final accuracy is not highly sensitive to $\rho_1 \text{ and } \rho_2$, as long as they are defined in a reasonable range. Importantly, the optimality of the obtained codebooks is not affected because in each iteration we are able to solve $\mathcal{IP}2$ optimally, thanks to the graph-coloring bound (Theorem 3). In the experimental study, we will provide discussion on how to choose these parameters including their appropriate ranges and their specific values for different $k$.
>
> 3) In line 127, with $L \approx k$, is assumed, to simply show that the number of  bi-linear terms  grows cubically with $k$ in $\mathcal{IP}1$. A more clearer way to state this would be that if $L \sim \mathcal{O}(k)$, then the number of bilinear terms are order $\mathcal{O}(k^3)$.
>
> 4) We will include remarks on the computational complexity of Algorithm 1.
>
> 5) Yes, for this case $\kappa_{\tilde{\mathcal{M}}} = \kappa_{\mathcal{M}}$. Lemma 2 lower bounds the error-correcting property of $\tilde{\mathcal{M}}$, with the error-correcting property of $\mathcal{M}$, i.e. $\kappa_{\mathcal{M}} \le \kappa_{\tilde{\mathcal{M}}}$. Theorem 2 upper bounds the error-correcting property of $\tilde{\mathcal{M}}$. For the case when, $ \xi(\mathcal{G}\_{\mathcal{M}} ) \ge 5 $, we have $\kappa_{\tilde{\mathcal{M}}} \le \kappa_{\mathcal{M}}$. Since both the upper-bound and lower bound are equal, therefore, $\kappa_{\tilde{\mathcal{M}}} = \kappa_{\mathcal{M}}$.
>
> 6) For all classification experiments we have reported an average of 5 runs. We believe that the improvements over multi-class CNN, and other codebooks such as 1-vs-1, 1-vs-all and sparse would easily pass the statistical significance test. In particular on large class datasets in table 5-7, our improvement over multi-class CNNs is consistently around 1-2%. And consistent with the literature, sparse codebooks have bad performance due to their ternary nature.

---

### Author Response · Authors · 2022-08-09
**Thanks to all the reviewers.**

As the author-reviewer discussion period comes to an end, we sincerely thank all the reviewers for their detailed review and follow-up discussions. Thank You.

---

### Meta-Review · Area_Chair_sAoz · 2022-08-28

**Recommendation:** Accept
**Confidence:** Certain

**Metareview:**

The paper furthers the understanding of designing codebooks in the context of using Error correcting codes for Multi-class problems. As opposed to continuous relaxation, the state of the art, it advocates a graph colouring approach which finally yields an alternative to codebooks. To the best of understanding the methodology will be hard to apply for large number of classes. Though the experimental results only show marginal improvement,  the methods does improve upon the state of the art.


**Award:**

No

---

### Decision · Program_Chairs · 2022-09-14

Accept